# Functional gradients in the human lateral prefrontal cortex revealed by a comprehensive coordinate-based meta-analysis

Majd Abdallah[1,2]*, Gaston E Zanitti[1,2], Valentin Iovene[1,2], Demian Wassermann[1,2]*

[1]MIND team, Inria, CEA, Université Paris-Saclay, Palaiseau, France; [2]NeuroSpin, CEA, Université Paris-Saclay, Gif-sur-Yvette, France

**Abstract** The lateral prefrontal cortex (LPFC) of humans enables flexible goal-directed behavior. However, its functional organization remains actively debated after decades of research. Moreover, recent efforts aiming to map the LPFC through meta-analysis are limited, either in scope or in the inferred specificity of structure-function associations. These limitations are in part due to the limited expressiveness of commonly-used data analysis tools, which restricts the breadth and complexity of questions that can be expressed in a meta-analysis. Here, we adopt NeuroLang, a novel approach to more expressive meta-analysis based on probabilistic first-order logic programming, to infer the organizing principles of the LPFC from 14,371 neuroimaging studies. Our findings reveal a rostro-caudal and a dorsoventral gradient, respectively explaining the most and second most variance in meta-analytic connectivity across the LPFC. Moreover, we identify a unimodal-to-transmodal spectrum of coactivation patterns along with a concrete-to-abstract axis of structure-function associations extending from caudal to rostral regions of the LPFC. Finally, we infer inter-hemispheric asymmetries along the principal rostrocaudal gradient, identifying hemisphere-specific associations with topics of language, memory, response inhibition, and sensory processing. Overall, this study provides a comprehensive meta-analytic mapping of the LPFC, grounding future hypothesis generation on a quantitative overview of past findings.

*For correspondence:
majd.abdallah@inria.fr (MA);
demian.wassermann@inria.fr
(DW)

Competing interest: The authors declare that no competing interests exist.

## Editor's evaluation

A meta-analysis of over 14,000 fMRI studies revealed a principle rostral-caudal gradient in the lateral prefrontal cortex. This gradient reflected an internal/external axis, which helps to organize the LPFC's involvement in widespread processes from affect, to memory, to control, and action. This is an important contribution to the literature, particularly as a meta-analytic approach has not been applied to this axis of organization and can complement the limitations of single studies. The paper is strengthened by the authors addressing a potential bias in the analysis and drawing a clearer relationship to functional networks.

## Introduction

The lateral prefrontal cortex (LPFC) supports a wide variety of cognitive processes considered hallmark features of the human brain (*Fuster, 2015*; *Petrides, 2005*). Understanding the functional organization of the LPFC is thus important to studying adaptive human behavior. Yet, the overarching organizing principles of the LPFC are still actively debated, with a variety of proposals on whether it is unitary, hierarchical, or constitutes a set of essentially separable networks subserving distinct

functions (*Badre and D'Esposito, 2009*; *Duncan, 2010*; *Goulas et al., 2012*; *Nee and D'Esposito, 2016*; *Reynolds et al., 2012*; *Nee, 2021*). Recently, there have been a few large-scale attempts to map the entire LPFC using both conventional and meta-analytical approaches. So far, these mappings have lacked specificity partly due to the limited breadth and complexity of queries that widely used tools can express and solve. In this study, we adopt a novel approach to expressive neuroimaging meta-analysis based on symbolic artificial intelligence to infer the organizing principles of the LPFC from thousands of studies with enhanced specificity.

The versatility of the LPFC suggests that it is far from being a unitary brain structure (*de la Vega et al., 2018*; *Dixon et al., 2018*; *Goulas et al., 2012*; *Koechlin et al., 2003*; *Nee and D'Esposito, 2016*). An influential class of hypotheses emerging from the domain of abstraction and hierarchical cognitive control proposes a rostrocaudal gradient in the LPFC. In this spatial layout, caudal regions respond to immediate sensory stimuli, middle regions select actions based upon a prevailing context, and rostral regions integrate concrete representations into more abstract rules to enable top-down temporal control of behavior (*Azuar et al., 2014*; *Badre, 2008*; *Badre and D'Esposito, 2009*; *Botvinick, 2008*; *Christoff and Gabrieli, 2000*; *Fuster, 2015*; *Koechlin et al., 2003*; *Jeon and Friederici, 2013*; *Nee and D'Esposito, 2016*; *Petrides, 2005*). A second class of hypotheses holds that a dorsoventral functional gradient, separating regions involved in distinct stimulus domains, governs the distribution of functions in the LPFC (*Rahm et al., 2013*; *Petrides, 2005*; *Parlatini et al., 2017*). More recent results suggest that the functionally distinct ventral and dorsal LPFC are each organized along the rostrocaudal axis according to the level of abstraction in task representations (*Petrides, 2005*; *Bahlmann et al., 2015*; *Blumenfeld et al., 2013*).

Extensive evidence from systems neuroscience further suggests that the LPFC comprises separable functional networks sub-serving different roles. The networks are the dorsal and ventral attention networks, default mode network, salience network, and most importantly the frontoparietal cognitive control network (*Cole et al., 2014*; *Dixon et al., 2018*; *Fedorenko et al., 2011*; *Marek and Dosenbach, 2018*; *Yeo et al., 2011*). These networks are believed to be situated upon a brain-wide connectivity gradient wherein the transmodal regions of the default mode network are maximally distant from unimodal regions of the sensorimotor/attention networks (*Huntenburg et al., 2018*; *Margulies et al., 2016*), with the salience and frontoparietal control networks occupying intermediate zones. One proposal holds that this spatial principle ascribes the LPFC with the role of integrating concrete and abstract representations following an external/present-oriented to internal/future-oriented gradient extending outwardly from the motor cortex (*Nee, 2021*). However, studies of the causal interplay between LPFC regions argue against such a linear gradient and rather support the hypothesis of distinct networks interacting within global and local hierarchies (*de la Vega et al., 2018*; *Bzdok et al., 2016*; *Dixon et al., 2018*; *Reynolds et al., 2012*; *Nee and D'Esposito, 2016*; *Nee and D'Esposito, 2017*; also see *Badre and Nee, 2018* for a comprehensive review). Within this systems-based framework, the middle LPFC, rather than the rostral LPFC, is believed to act as the focal point that integrates concrete and abstract representations, with increasingly rostral and caudal LPFC regions acting in their specific domains (e.g. internally or externally focused cognition) (*Badre and Nee, 2018*; *Nee and D'Esposito, 2017*; *Nee, 2021*).

The LPFC is also characterized by inter-hemispheric functional asymmetries, most notably for language (*Abbott et al., 2010*; *Fedorenko et al., 2011*) and inhibitory control processes (*Garavan et al., 1999*; *Aron, 2007*). Functional asymmetries between hemispheres are believed to arise from dynamic patterns of inter- and intra-hemispheric connectivity that represent organizing principles of functional specializations whose putative role is to promote efficient control of behavior (*Hartwigsen et al., 2019*; *Gonzalez Alam et al., 2021*). Thus, mapping the LPFC must take into account differences across hemispheres, especially in the distribution of lateralized functional associations. While there is a preponderance of research on the organization of the left LPFC in the fields of hierarchical control and language (*Nee, 2021*), a comprehensive comparison is yet to draw firm conclusions regarding the specifc functional associations of both left and right LPFC.

The multitude of proposals on the LPFC organization mainly arises from the diversity of protocols and researchers' degrees-of-freedom across studies. Thus, the idiosyncrasies of individual fMRI studies (e.g. task type, timing, magnitude of stimuli/responses, data analysis methods, and publication bias) can limit generalizability (*Botvinik-Nezer et al., 2020*; *Jennings and Van Horn, 2012*). And besides concerns of small sample sizes (*Turner et al., 2018*), each individual study probes a narrow

scope of a wide range of functions that putatively engage the LPFC, posing a risk of interpreting the results based on a small set of task contrasts. Therefore, it remains unclear to what extent the functional boundaries derived from each individual study correspond to the gross organization of the LPFC. Ultimately, a comprehensive meta-analysis is needed to synthesize the findings and infer consensus on the organization of the LPFC. Unlike individual studies, meta-analysis offers an over-arching perspective on brain activity, mapping a wide range of functions onto structures in a single statistical framework (*Fox et al., 2014*; *Müller et al., 2018*; *Yarkoni et al., 2011*).

Notwithstanding, the existing meta-analyses to map the LPFC, although informative, have been limited in scope and, more importantly, in tools. These limitations preclude a reliable distinction of closely related LPFC regions. On the one hand, due to the difficulty of the manual compilation of acti-vation peaks from the literature, most meta-analyses have been restricted to some regions (e.g. right inferior frontal gyrus *Hartwigsen et al., 2019*) or functions (e.g. working memory *Nee et al., 2013*). On the other hand, large-scale automated meta-analyses have assumed that LPFC regions are clusters of piece-wise constant coactivation, ignoring gradual functional transitions from one region to another (*de la Vega et al., 2018*). Finally, commonly used tools, such as Neurosynth (*Yarkoni et al., 2011*), are not expressive enough to represent complex hypotheses of specific functional associations in the LPFC. For example, it is arguably difficult to query a database on the probability that a function is associated with a study, given that it reports activation in one region and the simultaneous absence of activation in another region. This expressivity limitation becomes more challenging when performing a meta-analysis with multiple psychological concepts and regions that may coactivate across several tasks.

In this study, we overcome these challenges by using NeuroLang, a domain-specific programming language based on probabilistic first-order logic (*Iovene and Wassermann, 2020*). More specifi-cally, we perform a coordinate-based meta-analysis on 14,371 articles from the Neurosynth dataset (*Yarkoni et al., 2011*) along with a gradient-mapping approach (*Margulies et al., 2016*) to identify the organizing principles of activity in the LPFC. NeuroLang provides a structured and more expres-sive formalism to test neuroscience hypotheses and solve complex queries on large databases (*Iovene and Wassermann, 2020*). For instance, we can express queries of functional specificity in the likes of: *"What is the probability that empathy is associated with a given activation in the rostral LPFC and there does not exist any activation reported in caudal or middle LPFC?"*. Moreover, NeuroLang brings the power of probabilistic reasoning to account for uncertainty in heterogeneous data, such as in peak locations, region coactivations, and functional associations, all in a unifying framework. Importantly, a meta-analysis performed using NeuroLang is highly reproducible; the same queries used in this study can be used by future studies to validate or update the results as more data becomes available.

## Results
### A principal rostrocaudal and a secondary dorsoventral gradient explain most of the variance in meta-analytic connectivity in the LPFC

In the first analysis, we assess the extent to which LPFC regions are similar in their coactivation patterns across the literature. In other words, we want to identify the main gradients of meta-analytic connec-tivity in the LPFC (*Margulies et al., 2016*; *Huntenburg et al., 2018*). We start by reducing high-dimensional voxel-level data to lower-dimensional region-level data to increase interpretability and alleviate computational burdens. To do this, we project voxel activation data onto 1024 continuously-valued regions from the Dictionaries of Functional Modes (DiFuMo) atlases (*Dadi et al., 2020*). The DiFuMo is a set of probabilistic brain atlases created using data from thousands of subjects across 27 studies, comprising 2192 task-based and resting-state fMRI sessions publicly available on OpenNeuro (*Markiewicz et al., 2021*). Second, we write a program in NeuroLang that infers the probability for each DiFuMo brain region to be reported active given activation in each LPFC region and the proba-bility that a brain region is active given no activation in each LPFC region. We then compute the loga-rithm of the odds ratio (LOR) or the difference of the logits of the odds of these two events for each pair of regions. This yields a $N \times M$ matrix that encodes meta-analytic connectivity between $N$ LPFC regions and $M$ brain regions. To derive the gradients of meta-analytic connectivity in the LPFC, we apply a non-linear dimensionality reduction technique known as diffusion embedding (*Coifman et al., 2005*; *Margulies et al., 2016*) for the left and right LPFC separately. The resultant low-dimensional

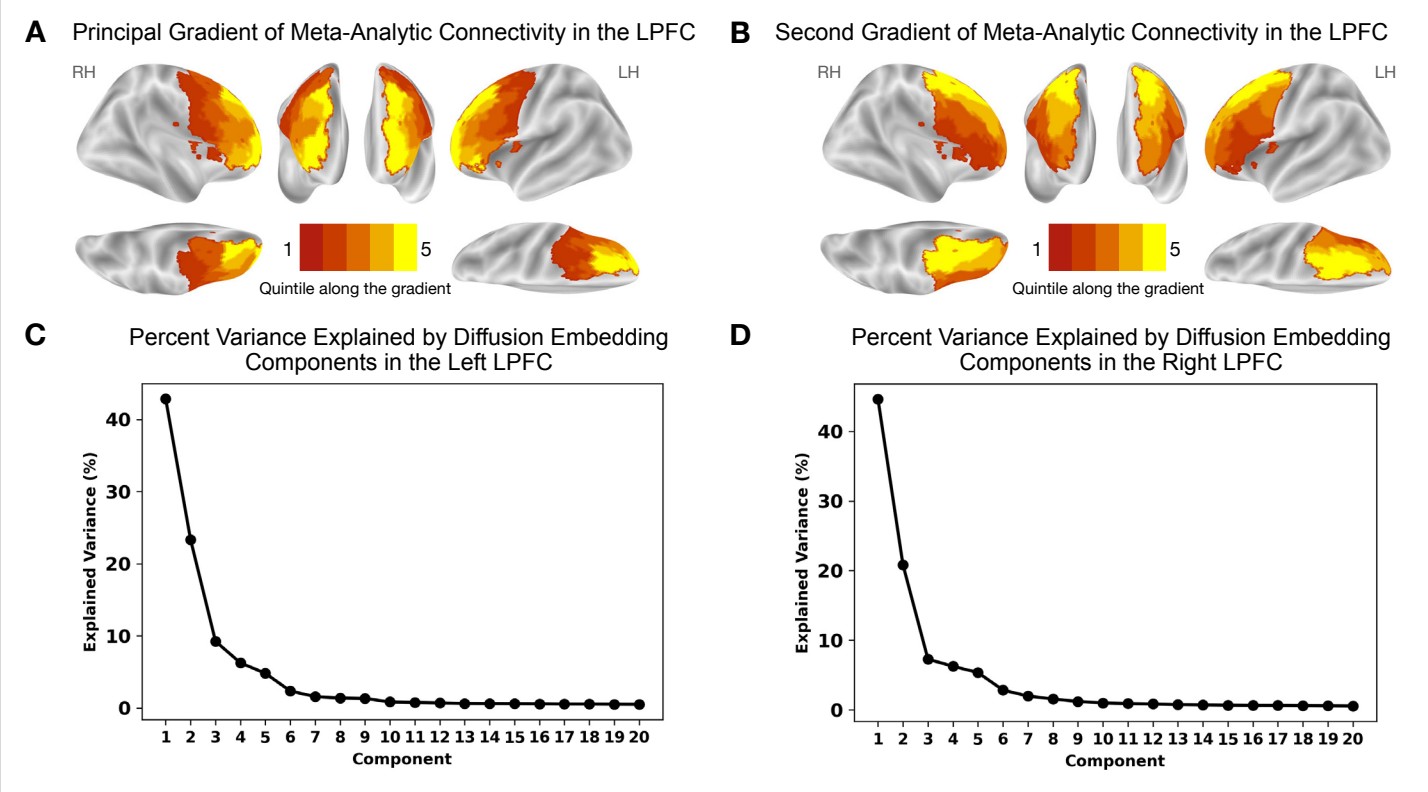

**Figure 1.** Meta-analytic connectivity gradients of the LPFC. The rostrocaudal and dorsoventral gradients explain the greatest amount of variance in meta-analytic connectivity in the LPFC. (**A**) The principal gradient in both hemispheres echoes a widely proposed rostrocaudal organization in the LPFC. This gradient represents the dominant direction of variations in connectivity patterns. (**B**) The gradient that explains the second-most variance in meta-analytic connectivity in the LPFC echoes a dorsoventral organization extending from ventrolateral to dorsolateral PFC regions. (**C**) and (**D**) The percentage of variance explained by the first 20 diffusion embedding components in the left and right LPFC, respectively.

The online version of this article includes the following figure supplement(s) for figure 1:

**Figure supplement 1.** The spatial layout of the principal LPFC gradient across 5000 re-runs of the meta-analysis on random sub-samples of the Neurosynth dataset.

**Figure supplement 2.** Percentage of variance explained by diffusion embedding components across 5000 re-runs of the meta-analysis.

**Figure supplement 3.** The principal LPFC gradient of long range coactivations.

**Figure supplement 4.** The principal LPFC gradient at the single-subject level.

**Figure supplement 5.** Spatial correlation between the subject-level and literature-level LPFC gradients.

embeddings situate each LPFC region along multiple axes, also known as gradients, each representing a direction of gradual variation in connectivity and explaining a proportion of the variance. The axis that accounts for the greatest amount of variance in meta-analytic connectivity is called the principal gradient, which will be the focus of the rest of the study.

Results are depicted in *Figure 1* and *Figure 2*. *Figure 1A* shows the principal gradient of meta-analytic connectivity in the LPFC. This gradient is anchored at one end by caudal LPFC regions and the other end by rostral LPFC regions, supporting a dominant rostrocaudal organization in the LPFC. The rostrocaudal spatial layout of the principal gradient is clearly expressed in terms of the posterior-to-anterior and inferior-to-superior positions of brain regions grouped in successive twenty-percentile gradient segments in *Figure 2*. One interesting question is whether the principal LPFC gradient is a local processing stream within the macroscale gradients described by *Margulies et al., 2016*, like the first gradient (separates sensimotor from default mode regions) and the third gradient (separates task-positive regions from the default mode network), or it represents a region-specific gradient. Therefore, within an LPFC mask, we compare the spatial layout of principal LPFC gradient (*Figure 2—figure supplement 1*) with that of the macroscale resting-state gradients from *Margulies et al., 2016*, namely gradient 1 and gradient 3. We observe a moderate positive correlation with gradient 1, and a

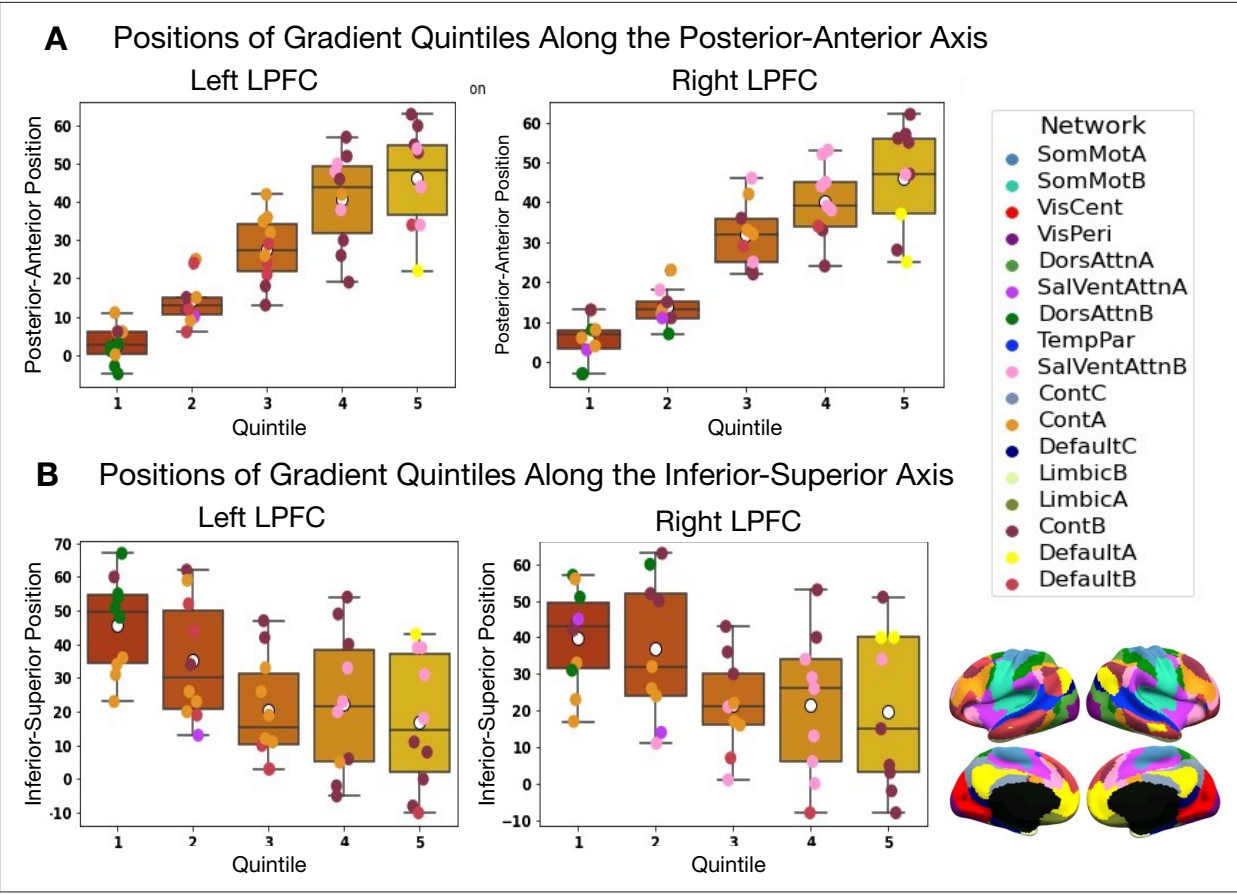

**Figure 2.** Positions along the posterior-to-anterior and inferior-to-superior axes in MNI space of quintile bins along the principal LPFC gradient. (**A**) Positions of quintile bins in the left LPFC. (**B**) Positions of quintile bins in the right LPFC. Each colored sphere represents a brain region, with the color reflecting its network membership within the 17-Networks atlas (**Yeo et al., 2011**). SomMot: SomatoMotor, VisCent/Peri: Visual Central/Peripheral, SalVentAttn: Salience/Ventral Attention, DorsAttn: Dorsal Attention, TempPar: Temporo-Parietal, Cont: Executive Control, Default: Default Mode.

The online version of this article includes the following figure supplement(s) for figure 2:

**Figure supplement 1.** Spatial correlation between the meta-analytic and resting-state connectivity gradients in the LPFC.

weak negative correlation with gradient 3, in both the left and right LPFC. These data show that the distribution of activity in the LPFC is dominated by a rostrocaudal gradient that fits within a more global gradient separating sensorimotor systems from higher-order association regions often implicated in abstract mental functions (**Margulies et al., 2016**; **Huntenburg et al., 2018**). Moreover, the principal gradient seems to be robust to different methodological choices, such as the choice of studies to be included in the meta-analysis (**Figure 1—figure supplement 1** and **Figure 1—figure supplement 2**) and the choice of coactivation distances between regions (**Figure 1—figure supplement 3**). Interestingly, the spatial layout of the principal LPFC gradient can be reproduced at the single-subject level, albeit individual-specific variations are observable (**Figure 1—figure supplement 4** and **Figure 1—figure supplement 5**). For a detailed description of the supplementary results and methods, please refer to Appendix 1. On the other hand, **Figure 1B** shows the secondary gradient of meta-analytic connectivity in the LPFC. This gradient extends along the dorsoventral axis of the LPFC, anchored at one end by ventral LPFC regions and the other end by dorsal LPFC regions. The topographies of the estimated gradients are consistent with those described in previous literature (**Petrides, 2005**; **Nee and D'Esposito, 2016**; **Bahlmann et al., 2015**). However, the extent to which each gradient explains the variation of activity in the LPFC across different brain states has remained unclear.

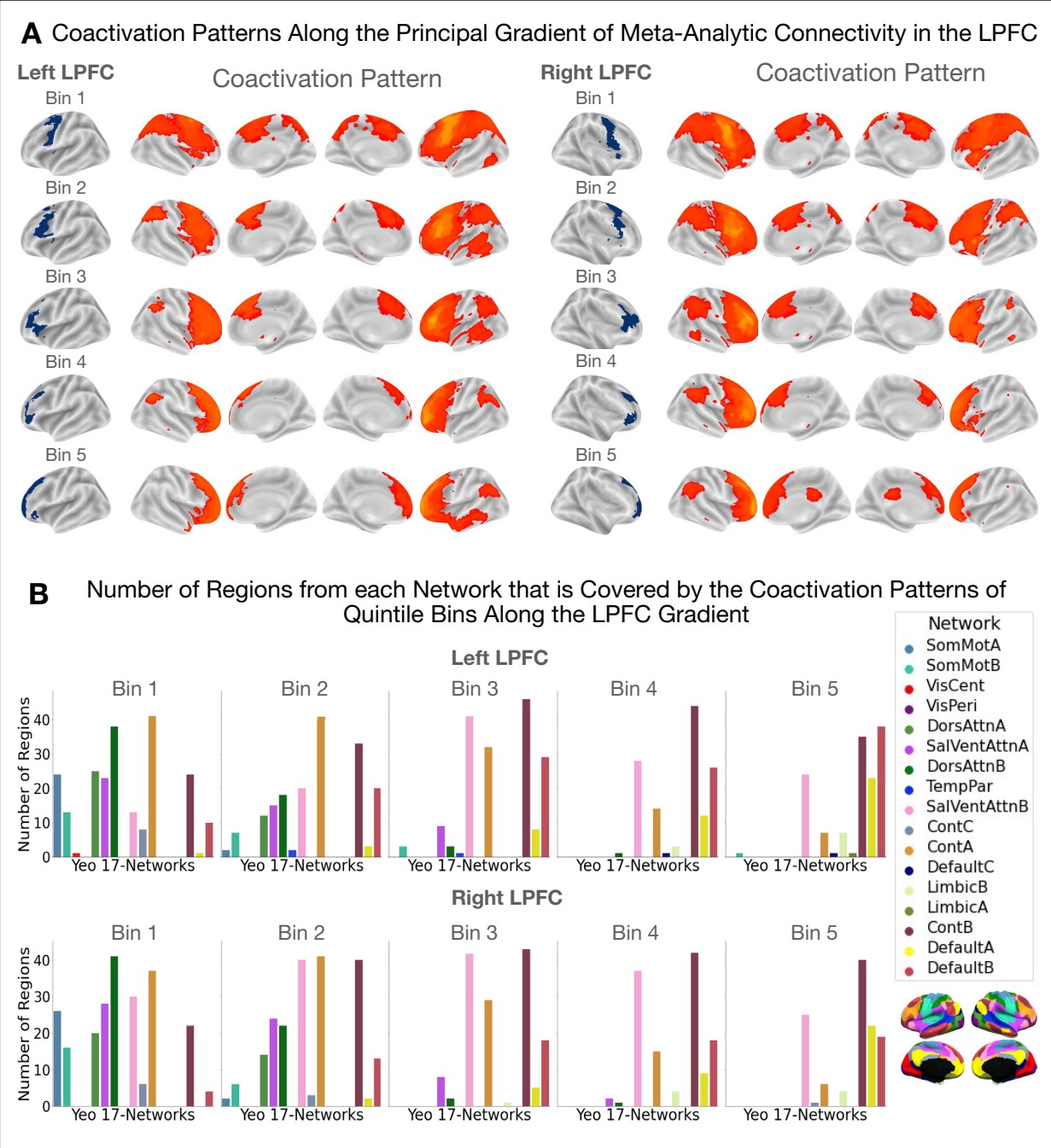

**Figure 3.** Coactivation patterns along the principal LPFC gradient. The coactivation patterns of quintile bins along the principal gradient in the LPFC capture a unimodal-to-transmodal spatial layout in brain network connectivity. (**A**) Coactivation patterns along the principal gradient in the left and right LPFC. Each brain map shows the regions that have a least three times the odds of being reported active given activation in a quintile bin relative to being active when activation is not reported in the quintile bin. Note that cerebellar and sub-cortical regions, although included in the analysis, are not shown in the figures. (**B**) Bar plots showing the number of regions from each network that overlaps with the coactivation pattern of each quintile bin. The data shown here suggests that the dorsal attention (green) and sensorimotor networks (blue) coactivate with the caudal bins (i.e. bins 1 and 2) more than with more rostral bins. On the other hand, the default mode network coactivates more with the rostral bins (i.e. bins 4 and 5) than with caudal bins.

## Coactivation patterns along the rostrocaudal axis of the LPFC follow a unimodal-to-transmodal organization

In the second main analysis, we characterize the principal LPFC gradient in terms of varying coactivation patterns of successive quintile bins and their overlap with canonical brain networks (*Figure 3*).

For this purpose, we write a NeuroLang program that first applies a smoothing spherical kernel with a 10mm radius around each peak in each study (*Wager et al., 2007*). The resulting binary activation maps (one map per study) are then projected onto 1024 functional regions defined by the DiFuMo atlas. However, this is not the case for quintile bins, where activation is considered reported in a quintile bin when at least one voxel is reported active within or near its vicinity ($< 3mm$). Finally, we estimate the LOR of the event that a brain region is reported active given activation in a quintile bin to the event that a brain region is active when no activation is reported in the bin.

Results are shown in *Figure 3*. *Figure 3A* depicts cortical coactivation maps of regions that exhibit at least threefold the odds (or $LOR > 0.5$) of coactivating with a bin than not. *Figure 3B* depicts bar plots showing the number of regions from each network that falls within each quintile bin's coactivation pattern. We observe a change in network connectivity profiles from a pattern dominated by unimodal dorsal attention/sensorimotor regions to one where overlap with the transmodal default mode regions becomes increasingly visible. In the middle zones, coactivation patterns overlap mostly with the salience and cognitive control networks that collectively are known to be part of the 'multiple demand network' (*Duncan, 2010*). These data support recent proposals of domain-general processing in the mid-LPFC regions as opposed to more domain-specificity at the extremities of the rostrocaudal LPFC gradient (*Nee, 2021*). Here, 'domain-specific' denotes a region's involvement in either internal/present-oriented or external/future-oriented processing, whereas 'domain-general' is used to indicate possible a region's involvement in both types of processes.

## Mapping specific topic associations in the LPFC supports the hypothesis of increasing abstract representations extending along the rostrocaudal gradient

In the third analysis, we characterize the principal LPFC gradient in terms of associations with 38 topics chosen from an original set of 100 topics (version-5 of topic modelling from Neurosynth) (*Poldrack et al., 2012*). These topics are data-driven aggregations of single terms that co-occur in article abstracts and cover broad cognitive and behavioral domains often studied in cognitive neuroscience. Refer to *Table 1* to see the five strongest loading terms on each topic listed in descending order of association strength. We perform the topic-based decoding using what we call 'segregation queries'. A segregation query infers the probability *"that a topic is associated with activation in a set of regions given the absence of activation in another set of regions"*. Expressing segregation queries using NeuroLang is straightforward and enables inferring more specific structure-functions relationships. In conventional non-segregation inferences, these associations may be blurred due to regions coactivating across many studies and tasks (*Figure 4—figure supplement 1*). The most probable reason for this blurring is that typical fMRI task contrasts rarely isolate regions underlying distinct but related processes, which likely need to be probed across multiple tasks to ensure the independence of regions (*Poldrack et al., 2011*).

The NeuroLang program of this analysis infers the probability that a topic is present given activation between bin a ($a \in [1, 5]$) and a bin b ($b \in [1, 5]$), while there exists no activation in any bin outside the range $[a, b]$ . The term 'there exists no' is the segregation expression that guarantees more specificity of associations. For instance, in the event where $a = 1$ and $b = 4$, the program queries the database on the probability that a topic is present in a study given activation of bins 1 and 4 (or any region in between) and given no activation outside the quintile range $[1, 4]$. In the event where $a = b$, we infer topic association given activation constrained in only one quintile bin. Concurrently, the program infers the probability of the opposite event by selecting the studies that do not match the criteria imposed by the segregation query. By computing the LOR of these two events, we obtain a measure of the evidence in favor of association between each topic and spatially constrained activation patterns along the principal gradient of the LPFC.

Results are depicted in *Figure 4A* and *Figure 4B* for the left and right LPFC, respectively. Topics are vertically ordered from top-to-bottom by the weighted mean of their location along the gradient. We observe a broad shift in topic associations from topics of sensorimotor/attention processing at more caudal regions to topics of emotion and memory-related topics at more rostral regions of the principal LPFC gradient. Between these extremities, we mainly observe topics related to executive functions and language. This pattern of topic-bin associations suggests that as activation extends away from the posterior towards the anterior LPFC, task representations become more abstracted

**Table 1.** Thirty-eight topics from the Neurosynth LDA-driven 100 topics set and the top five terms loading on each topic listed in descending order of association strength.

| topics | terms |
| --- | --- |
| Action Imitation | mirror; observation; imitation; action; gestures |
| Attention | attention; attentional; task; visual; control |
| Auditory Processing | auditory; sounds; sound; cortex; temporal |
| Cognitive Control | task; cognitive; performance; control; executive |
| Conflict Interference | conflict; interference; control; incongruent; congruent |
| Cued Attention | spatial; cues; cue; location; attention; orienting |
| Decision Making | decision; making; risk; choice; decisions; choices |
| Declarative Memory | memory; retrieval; episodic; mtl; memories |
| Emotion Regulation | regulation; emotion; reappraisal; cognitive; amygdala |
| Emotional Valence | emotional; negative; positive; amygdala; emotion |
| Empathy | social; empathy; moral; game; people |
| Eye Movements | eye; gaze; saccade; movements; target |
| Face/Affective Processing | amygdala; emotional; faces; facial; emotion; expression |
| Feedback/Error Processing | feedback; error; learning; errors; prediction |
| Judgement | judgments; judgment; ppc; reference; drawing |
| Lexical Semantics | semantic; words; word; lexical; verbs |
| Memory Encoding | memory; encoding; hippocampus; hippocampal; retrieval |
| Memory Retrieval | memory; recognition; items; retrieval; recollection |
| Mental Imagery | imagery; mental; events; future; imagined |
| Mentalizing | reasoning; mind; mental; social; tom |
| Motor Movement | motor; movement; cortex; movements; hand |
| Multisensory Processing | visual; motion; auditory; modality; sensory integration |
| Navigation | ms; spatial; virtual; navigation; illusion |
| Numerical Cognition | number; numerical; numbers; magnitude; size |
| Perception | perceptual; perception; interaction; sensory; visual |
| Reading | reading; words; word; phonological; chinese |
| Response Inhibition | inhibition; response; control; inhibitory; task |
| Response Selection | response; stimulus; trials; trial; presented |
| Reward Processing | reward; striatum; ventral; anticipation; monetary |
| Sentence Comprehension | language; comprehension; sentences; sentence; syntax |
| Skill Learning | learning; training; sequence; performance; practice |
| Somatosensory Processing | somatosensory; stimulation; tactile; hand; cortex |
| Speech Production | speech; auditory; production; perception; temporal |
| Subjective Experience | pictures; images; aversive; neutral; unpleasant |
| Target Detection | target; detection; targets; awareness; presented |
| Task Switching | task; switching; rule; set; switch |
| Visual Perception | object; objects; visual; category; cortex |
| Working Memory | memory; working; wm; task; load; verbal |

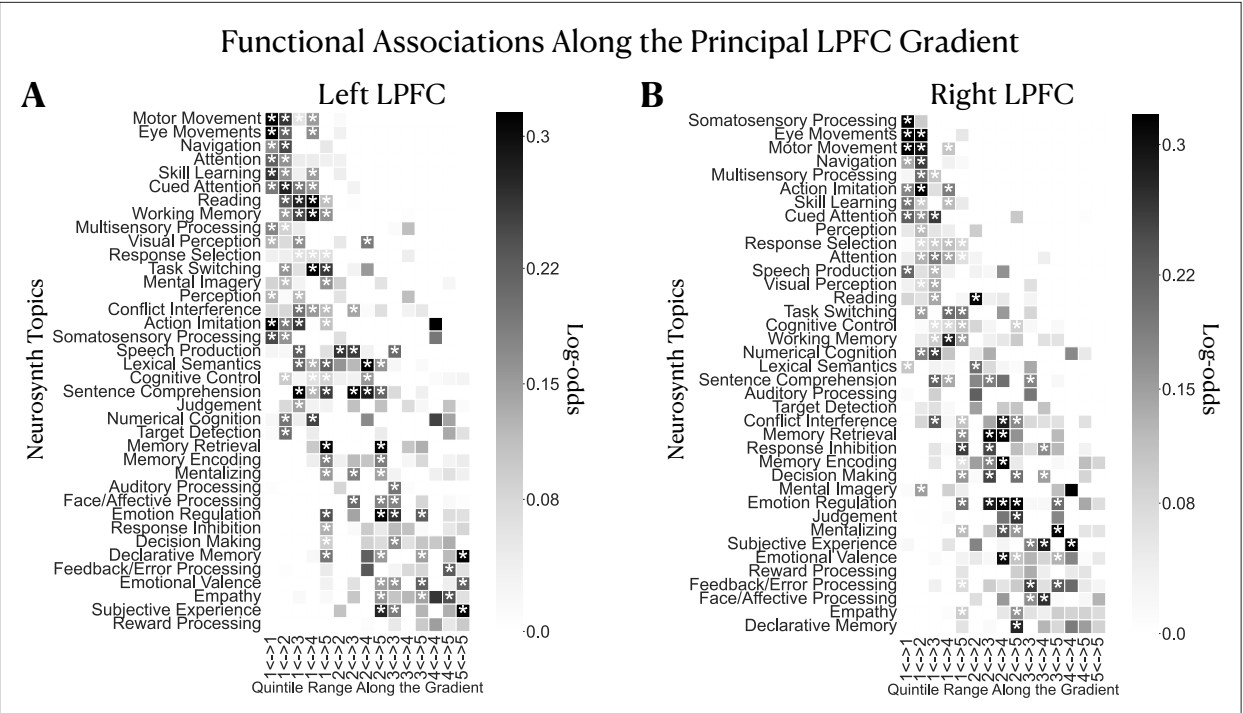

**Figure 4.** Inferring functional associations along the principal LPFC gradient using segregation queries. Mapping functional associations using segregation queries reveals a structured ordering of topics along the principal gradient in the (**A**) left and (**B**) right LPFC. Topics are ordered by the weighted mean of their location along the principal gradient. Topics of sensorimotor processing are mostly located at the top followed by executive functions and language, and finally emotion/memory/social cognition-related topics mostly occupy the bottom. Note that although the order of topics varies between hemispheres, the general profile of topic associations is comparable. A two-headed arrow along the horizontal axis signifies a coactivation constricted in a given range of quintile bins. Log-odds ratio values are thresholded at 0 to show only positive associations. White asterisks denote the log-odds ratio values whose 95% confidence interval estimated from 5000 re-runs of the meta-analysis on random sub-samples of the Neurosynth dataset does not include the value 0.

The online version of this article includes the following figure supplement(s) for figure 4:

**Figure supplement 1.** Inferring functional associations along the principal LPFC gradient without segregation queries.

from immediate demands of perception/action cycles. Finally, for the purpose of comparison with conventional decoding, we perform this analysis without using segregation queries. Here, we infer the probability that *"a topic is present in a study given activation reported in a set of regions"*. Results of the non-segregation-based analysis are shown in *Figure 4—figure supplement 1* and show an ordering of topics along a roughly concrete-to-abstract axis. However, the individual associations are rather distributed across bins, suggesting no specificity of structure-function associations in the LPFC.

## Gradient-based meta-analysis of inter-hemispheric asymmetries reveals lateralized associations with topics of language, memory, inhibitory control, sensory and error processing

The last analysis of this study aims at contrasting the two LPFC hemispheres in terms of specific topic associations in a gradient-like fashion. More precisely, we compare homologous quintile bins in both hemispheres in terms of functional associations. For this purpose, we write a NeuroLang program that solves segregation queries between hemispheres. Here, we infer the probabilities *"that a topic is present in a study given activation in a quintile bin in the right (respectively left) LPFC and there exists no reported activation in the entire left (respectively right) LPFC"*. The LOR, in this case, represents the amount of evidence for topic association given unilateral activation in the right hemisphere relative to a unilateral activation in the left hemisphere of the LPFC across quintile bins of the principal gradient.

Results are depicted in *Figure 5*. In general, we do not observe any systematic variation in the degree and nature of functional asymmetries when moving along the rostrocaudal gradient. The amount of evidence for hemispheric preference as well as the domains of lateralized topic associations

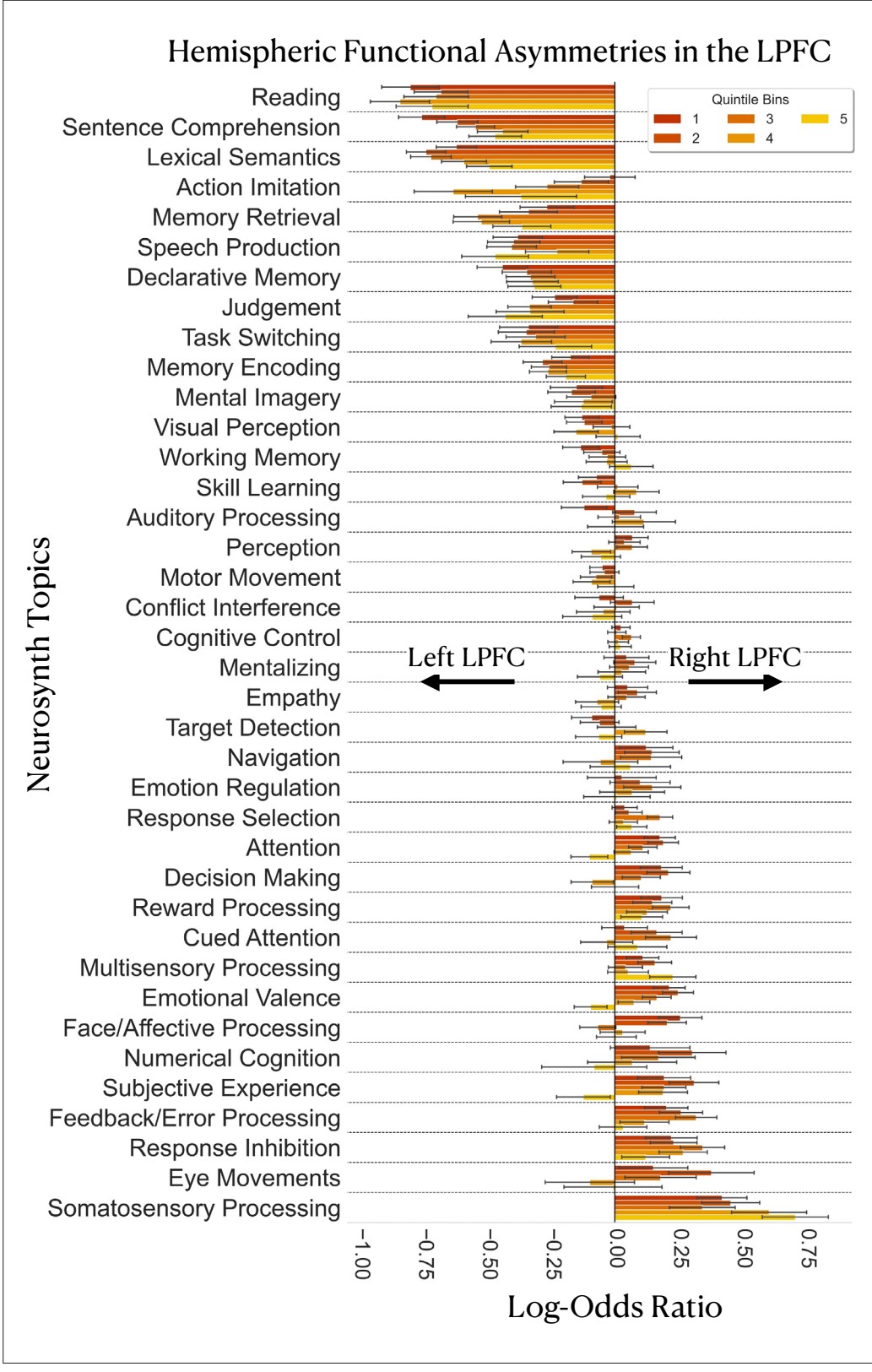

**Figure 5.** Gradient-based mapping of hemispheric asymmetries in the LPFC. Meta-analysis of inter-hemispheric asymmetries reasserts the left-hemispheric dominance of language and memory and the right-hemispheric dominance of inhibition and sensory processing/monitoring in the LPFC. Positive log-odds ratios indicate evidence in favor of right-hemispheric preference of a topic in a given bin, whereas negative values indicate evidence in

*Figure 5 continued on next page*

*Figure 5 continued*

favor of left-hemispheric preference of a topic in a given bin. Error bars represent the 95% confidence intervals estimated from 5000 re-runs of the meta-analysis on random sub-samples of the Neurosynth dataset. Each random sub-sample comprises 60% of the studies of the original dataset (around 8623 studies). Topics are ordered from most-left dominant to most-right dominant based on the average of the log-odds ratio values over the five quintile bins.

in the LPFC are comparable between caudal and rostral LPFC regions in both hemispheres (*Figure 5*). Specifically, we find that language-related topics, such as 'lexical semantics', 'sentence comprehension', and 'reading' are more likely to be associated with left LPFC activation from bins 1–5. Likewise, we find that memory-related topics, 'memory retrieval' and 'declarative memory', also show substantial evidence for left-hemispheric preference across multiple quintile bins of the principal LPFC gradient. In contrast, topics such as 'somatosensory processing', 'eye movements', 'response inhibition', 'feedback/error processing' are more likely to be associated with right LPFC activation across multiple quintile bins. Together, these results reassert the views on hemispheric functional asymmetries in the LPFC, with the left LPFC being involved in language and memory processes (*Abbott et al., 2010*; *Fedorenko et al., 2011*; *Gabrieli et al., 1998*; *Hartwigsen et al., 2019*) and the right LPFC is involved in stimulus-driven action control and sensory monitoring processes (*Aron, 2007*; *Bartolomeo and Seidel Malkinson, 2019*; *Thiebaut de Schotten et al., 2011*; *Garavan et al., 1999*).

## Discussion

In this study, we infer the LPFC gradients via a comprehensive meta-analysis. We find a principal rostrocaudal and secondary dorsoventral gradient that explain most of the variance in meta-analytic connectivity patterns in the LPFC across 14,371 studies. We also find a systematic distribution of network coactivation patterns along the principal gradient from regions putatively associated with external/present-oriented functions to regions of higher order networks associated with internal/temporally distant cognition. Finally, by assessing inter-hemispheric functional asymmetries along the principal gradient, we find patterns of lateralized topic associations consistent with previous findings on language, memory, sensory processing/monitoring, and response inhibition. Overall, the specificity of findings in this study grounds future hypothesis generation on a quantitative overview of previously published results.

### The principal meta-analytic gradient in the LPFC reflects a rostrocaudal organization characterized by domain-generality in the intermediate zone and domain-specificity at the extremities

The principal gradient of meta-analytic coactivation in the LPFC echoes a rostrocaudal organization in the sense that successive quintile bins along the gradient show a linear increase in their posterior—to-anterior position from the vicinity of the motor cortex toward the anterior of the brain. Thus, this gradient places caudal LPFC regions at the farthest point from rostral LPFC regions on a spectrum of similarity in meta-analytic connectivity patterns. This result agrees with a popular class of hypotheses emerging from abstraction and hierarchical control studies on a rostrocaudal gradient in the LPFC (*Badre, 2008*; *Badre and D'Esposito, 2009*; *de la Vega et al., 2018*; *Koechlin et al., 2003*; *Wendelken et al., 2012*). Yet, a question remains open concerning the properties and functional roles of different zones in the rostrocaudal LPFC gradient.

Early studies on the matter have ascribed the rostral LPFC with the roles of integrating concrete information from caudal regions and relaying back top-down control signals (*Koechlin et al., 2003*; *Badre, 2008*; *Petrides, 2005*; *Wendelken et al., 2012*). In contrast, recent studies relying on causal evidence argue against a unidimensional linear gradient and rather place the mid-LPFC as the nexus of both concrete and abstract representations (*Nee and D'Esposito, 2016*; *Badre and Nee, 2018*; *Nee, 2021*). Although we cannot infer such integrative processing by means of causality in the current study, we find that the mid-LPFC regions are situated in the intermediate zones of the principal gradient. Technically speaking, this means that the whole-brain coactivation profile of the mid-LPFC is not totally similar nor dissimilar to those of the caudal and rostral LPFC, but somewhat overlaps with them. Indeed, we observe that intermediate gradient zones coactivate more with the salience

(SalVentAttnB) and control networks (ContA and ContB) and to a lesser extent with both the attention (DorsAttnA and DorsAttnB) and default mode networks (DefaultA and DefaultB). This pattern is not observed for the two extremities of the gradient (bins 1 and 5), where networks involved either in external processing (SomMotA, DorsAttnA, and SalVentAttnA) or internal cognition (DefaultA and DefaultB) are respectively more dominant along with the salience and cognitive control networks (*Figure 3*). The salience and control networks are believed to be integrative and to mediate the interaction of the default and attention networks to control the transition between external/present-focused and internal/temporally-distant processing (*Menon and Uddin, 2010*; *Nee, 2021*). Finally, as further support, when activation is restricted within the caudal-to-middle zones of the gradient (bins 1–4), we mainly observe associations with topics of action execution, sensory perception, language, and executive control (*Figure 4*). In contrast, when coactivation is restricted within the middle-to-rostral zones (bins 2–5), we observe associations with topics of memory, emotion, and social cognition—functions that rely on abstract representations untethered from immediate environmental demands (*Buckner and Krienen, 2013*). These patterns of network and function associations support a domain-general role of mid-LPFC regions and a more domain-specific role (i.e. either internally or externally oriented processing) for caudal and rostral LPFC regions.

## The principal gradient of meta-analytic connectivity in the LPFC echoes the principle gradient of brain-wide intrinsic connectivity

As seen in *Figure 2—figure supplement 1*, the topography of the principal LPFC gradient correlates with the overall layout of the principal brain-wide gradient described in *Margulies et al., 2016*, which represents the dominant spatial principle governing resting-state connectivity throughout the entire cerebral cortex. This spatial principle conceptualizes higher order cognition as emerging from dynamic interactions of large-scale networks, systematically organized along an axis of abstraction that extends from unimodal sensorimotor regions to transmodal default mode regions (*Mesulam, 1998*; *Margulies et al., 2016*; *Huntenburg et al., 2018*). Importantly, it incorporates the seemingly isolated local processing streams across the cortex within a continuous global framework. In this sense, the spatial location of a brain region is not arbitrary; a region's position along the principal gradient is a major determinant of its connectivity profile, its network membership and consequently its functional role. Specifically, it has been found that the longer the spatial distance between a region and the primary cortices, the more distant are its functional connections and the more it is dispositioned to subserve abstract mental functions (*Oligschläger et al., 2017*). The default mode network occupies the top end of the global intrinsic connectivity gradient and exhibits the greatest geodesic distance from the sensorimotor cortices, allowing it to process highly internalized information abstracted from immediate sensory input (*Margulies et al., 2016*; *Smallwood et al., 2021*). In a similar sense, we find that the rostrocaudal gradient in the LPFC captures systematic transitions in large-scale functional networks (*Figure 2* and *Figure 3*), such that caudal zones pre-dominantly coactivate with sensorimotor/attention networks, middle zones coactivate with salience/executive control networks, and increasingly rostral gradient zones coactivate more with the default mode network. This result confers a spectrum of increasing abstraction on the LPFC that follows the transition from unimodal to transmodal regions.

Further supporting this view are the specific topic associations along the principal LPFC gradient inferred using segregation queries (*Figure 4*). These patterns of functional associations are not evident from the non-segregation-based topic association analysis, although the topics seem to follow a roughly concrete-to-abstract ordering. As shown in *Figure 4—figure supplement 1*, topic associations are either equal to zero for all bins or cover the entire LPFC gradient, suggesting no specificity in structure-function associations. Thus, our results stress the importance of using segregation queries to infer relatively specific associations often masked in conventional functional decoding analyses. Overall, the rostrocaudal LPFC gradient described herein represents a literature-inferred map of a concrete-to-abstract organizing principle, wherein globally interacting networks interface locally in the LPFC to support adaptive behavior within dynamic contexts (*Badre and Nee, 2018*; *Nee and D'Esposito, 2016*).

## Meta-analysis of LPFC inter-hemispheric asymmetries reasserts the left-hemispheric preference of language and memory processes and the right-hemispheric preference of inhibitory and sensory processes

Segregation-based meta-analysis of inter-hemispheric asymmetries reveals associations with language, memory, response inhibition, error-processing, and somatosensory processing in the LPFC. The importance of segregation queries, in this case, is in inferring the structure-function associations whose presence is only predicted by unilateral activation in the LPFC. Previously, the lateralization of function in the brain has been well documented for certain functions, notably language (*Abbott et al., 2010*; *Fedorenko et al., 2011*) and response inhibition (*Aron, 2007*). More recently, an effort to map hemisphere-specific functions across the whole brain (*Karolis et al., 2019*) has uncovered four global dimensions of laterlization: symbolic communication, perception and action, emotion, and decision making. However, a comprehensive comparison of hemisphere-specific functional associations within the LPFC remains lacking, especially when taking into account the principal organizing gradient in each hemisphere. The analysis carried out in this study is one step forward toward filling this gap.

In general, we do not observe any systematic variations in the degree nor nature of lateralized topic associations along the principal LPFC gradient (*Figure 5*). That is, unilateral structure-function association patterns seem to be comparable throughout the rostrocaudal gradient. For instance, the greatest observed evidence for left-hemispheric preference in the LPFC is attributed to language and memory-related topics, which is consistent with a long line of research on the linguistic and semantic selectivity of the left hemisphere (*Gonzalez Alam et al., 2021*). In contrast, the greatest amount of evidence for right-hemispheric preference in the LPFC is attributed to 'response inhibition' and 'somatosensory processing', and to a lesser extent 'feedback/error processing' and 'eye movements'. These results are consistent with the role of the right hemisphere, in general, in sensory monitoring and the inhibitory processes. Notwithstanding, we observe relatively weak evidence (*Figure 5*) for right-hemispheric preference of attention-related topics, such as 'attention', 'cued attention', and 'navigation', although such topics are often attributed to the right brain hemisphere (*Bartolomeo and Seidel Malkinson, 2019*; *Thiebaut de Schotten et al., 2011*). While there may be more than one explanation for these observations, a plausible one is related to the data-driven nature of topics. More specifically, given that topics are 'bags' of words that frequently co-occur in the abstracts of articles, they are at best proxies to the actual mental functions. This means that topics can not be specific enough to capture finely grained cognitive constructs. Nonetheless, topics are relatively better representatives of psychological domains than individual terms that pose the risk of being interpreted out of context (*Poldrack et al., 2012*). Overall, the current findings support the preferential roles of the left LPFC in language, semantics, and memory processes and the right LPFC in sensory monitoring and cued inhibition of behavior.

### Limitations

While the present results provide a relatively unbiased mapping of the organizing gradients in the LPFC through meta-analysis, several caveats and limitations are worth noting. First, although meta-analysis is arguably one of the best techniques to synthesize findings across the literature, it amplifies spatial uncertainty by combining a large number of different datasets and hence might overly smooth the gradients. In addition, the methods adopted here, such as the spatial smoothing prior (10mm radius) and the use of a brain atlas, may further exacerbate the problem and make the organization of the LPFC appear more spatially continuous than it actually might be. Indeed, the actual steps along the rostrocaudal axis might be discrete/areal, separating disparate networks as in clustering-based representations (e.g. Yeo 17 atlas), may approach total continuity, or something in between. Although the supplementary subject-level analysis shows that individual gradients are more granular than the literature-based gradient (*Figure 1—figure supplement 4*), we do not make any claims about the ground truth spatial resolution of the principal LPFC gradient, as no evidence has been observed to support any of the hypotheses.

Second, we make simplifying assumptions to alleviate computational burdens, notably the use of 1024 functional regions from the DiFuMo atlases (*Dadi et al., 2020*) and the choice of 20-percentile gradient bins as units of analysis. These assumptions might impose a fixed dimensionality and force voxels to be grouped in static regions, which ignores dynamics in brain activity observed at multiple timescales within and across individuals (*Salehi et al., 2020*). Another simplifying assumption is the use

of topics that represent broad concepts built upon the frequency with which simpler terms co-occur in studies, ignoring more finely-grained cognitive structures. Moreover, some of the simple terms may seemingly not fit within the most likely description of the topic. Integrating ontologies, such as the Cognitive Atlas (*Poldrack et al., 2011*), will arguably improve the ability of automated meta-analyses to differentiate fine-grained cognitive constructs. In fact, NeuroLang is well-equipped to integrate ontologies into meta-analyses, and this will be our next step toward improving the precision of meta-analytic queries.

Third, small sample sizes per study and potential publication bias, or the tendency of authors and journals to only publish positive and statistically significant results (*Jennings and Van Horn, 2012*), might impact the reliability of the current findings. Even though spatial smoothing priors and probabilistic brain atlases may alleviate some bias, future meta-analyses should rely on complete data like unthresholded statistical images stored in large repositories, such as NeuroVault (*Gorgolewski et al., 2015*), to validate the results. Finally, an important limitation, not specific to this meta-analysis, is that the current knowledge of task-dependent activations in the brain is as good as the task paradigms that induce these activations (*Poldrack and Yarkoni, 2016*). More broadly, an ongoing endeavor in cognitive neuroscience is developing the appropriate paradigms that isolate cognitive processes of closely related brain regions (*Poldrack and Yarkoni, 2016*). Studies in the domain of abstraction and hierarchical control use nested tasks classed by different levels of abstraction, which can uncover functional gradients in the LPFC (e.g. *Koechlin et al., 2003*; *Nee and D'Esposito, 2016*). However, these studies are not common in the literature and are limited to a small range of functions. In contrast, the bulk of tasks included in the Neurosynth dataset, while not hierarchical, captures a much wider variety of brain states, but at the expense of losing some level of specificity.

## Conclusion

In conclusion, the present study provides a meta-analytic mapping of the principal organizing gradients in the LPFC of humans. The LPFC appears to be organised along two spatial gradients, rostro-caudal and dorsoventral, that respectively explain the most and second-most variance in meta-analytic connectivity. We also find that the principal gradient captures a unimodal-to-transmodal spectrum of increasing abstraction in network connectivity and functional associations. Importantly, we overcome the limitations of previous large-scale attempts using a novel domain-specific query language, NeuroLang, to formulate expressive queries on the largest coordinate-based meta-analysis database to date. As more studies are aggregated into future databases, the analyses carried out in this study can be reproduced using the same queries as well as extended to explore other brain regions.

## Methods
### Data and software

We use the latest version of the Neurosynth dataset (*Yarkoni et al., 2011*) last updated in July 2018 to include 14,371 articles that include more than 500,000 activation coordinates covering the whole brain. Each study in the database is represented by a PubMed ID, peak activation coordinates and weighted topic associations. Neurosynth's peak activation coordinates are either reported in MNI space or Talairach space. However, we re-sampled all activation data to the symmetric 3 mm MNI template before any analysis took place. Moreover, all analysis are carried out in volumes, while surfaces are used for visualization only. To assess topic-associations in the LPFC, we use the set of 100 Neurosynth topic terms (version 5) previously generated by applying latent Dirichlet allocation to the abstracts of articles in the database (*Poldrack et al., 2012*). Out of the 100 topics, we include 38 topics (shown in *Table 1*) that represent coherent cognitive functions, excluding those that correspond to subject populations (e.g. brain disorders, age, sex), brain anatomy, imaging modalities and analysis techniques. All analyses and visualizations are implemented in python. In particular, we use the NeuroLang (https://github.com/NeuroLang/NeuroLang; *Wassermann et al., 2022*) library to perform all meta-analysis steps and the BrainSpace library (https://github.com/MICA-MNI/BrainSpace; *Vos de Wael et al., 2022*) to estimate low-dimensional embeddings of meta-analytic connectivity patterns in the LPFC (*Vos de Wael et al., 2020*). All source code and data files used in this study will be publicly available to be openly accessed on github at https://github.com/majdabd/lpfc_gradients-meta-analysis (copy archived at swh:1:rev:7ab4efcbc9875b92745b7b1c43864d7352fd3b90; *Abdallah, 2022*).

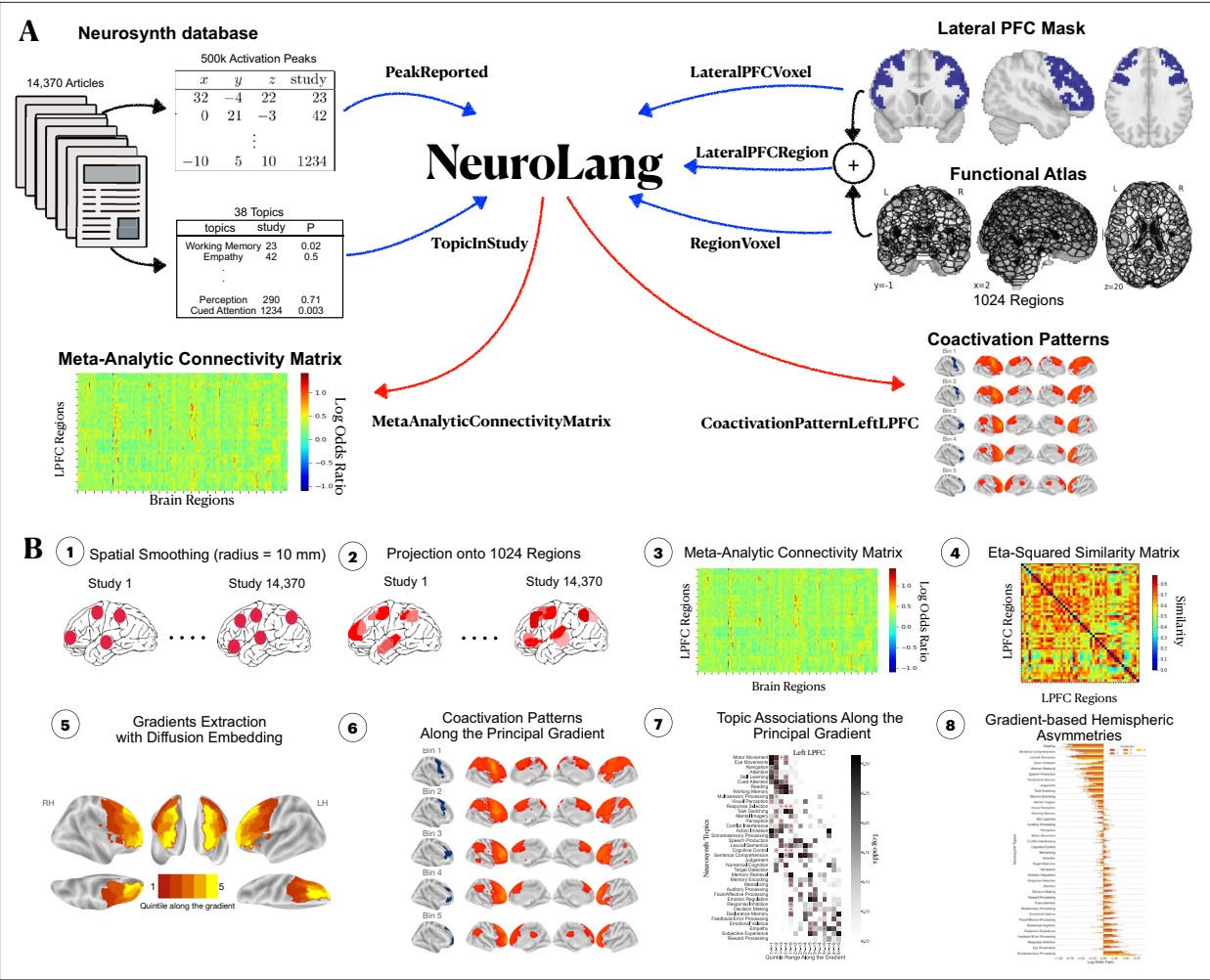

**Figure 6.** Schematic overview of our analysis pipeline. (**A**) Inputs and outputs of NeuroLang. Inputs are represented using blue arrows and include: Peak activations and topics from the Neurosynth dataset, the lateral PFC mask, and the 1024 regions from the DiFuMo atlases are represented in a unifying framework within NeuroLang. Two examples of outputs are shown here and represented using red arrows. (**B**) The main steps of the meta-analysis carried out in this study. (1) Spatial smoothing with 10 mm kernel around each peak. (2) The binary activation map of each study is projected onto 1024 functional regions. Varying shades of red signify that regions have different probabilities of being reported by a study depending on the location of voxels within each region. (3) The meta-analytic connectivity matrix encodes the log-odds ratios of coactivation between each region in the LPFC and every region in the brain. (4) A similarity matrix encodes the degree of correspondence between LPFC regions in their meta-analytic connectivity profiles, estimated by the eta-squared similarity metric. (5) The principal gradient of meta-analytic connectivity in each hemisphere is then derived from the similarity matrix using diffusion embedding. (6) Coactivation patterns of successive quintile gradient bins are inferred (7) Specific topic associations along the principal gradient are inferred using segregation queries. (8) Finally, a gradient-based meta-analysis of hemispheric asymmetries is performed.

## The lateral prefrontal cortex mask

To facilitate the selection of regions in the LPFC for meta-analysis, a spatial mask of the LPFC is needed. We rely on a previously created mask of the lateral frontal lobe created from *de la Vega et al., 2018*. However, we exclude voxels with less than 25% probability of falling in the grey matter as well as voxels located at $x < 18$ or $x > -18$ from the midline of the brain to ensure that regions in the anterior and superior parts of the medial prefrontal cortex are not included. We also exclude voxels in the orbitofrontal cortex and anterior insula, while making sure to include voxels of the lateral orbitofrontal cortex. Finally, to focus our analysis on the association regions of the lateral frontal lobe (i.e, the LPFC), we exclude voxels in the motor cortex as defined by the somatomotor networks of the 17-Networks atlas (*Yeo et al., 2011*). The LPFC mask is shown in *Figure 6A*.

## The 1024 functional regions dictionary from DiFuMo

To increase the interpretability of our findings and alleviate computational burdens, we reduce voxel-level data to region-level data. In particular, we adopt the 1024 functional regions dictionary from the Dictionaries of Functional Modes (DiFuMo) atlases (*Dadi et al., 2020*). The DiFuMo is a set of multi-scale functional atlases estimated via massive online dictionary learning (*Mensch et al., 2016*) applied to functional brain volumes of thousands of subjects across 27 large-scale studies, forming a total of 2192 task-based and resting-state MRI sessions. Reducing voxel data to 1024 functional regions has been argued to capture the functional neuroanatomy of the brain equally well as voxel-level analysis while reducing computational burdens (*Dadi et al., 2020*). Unlike other dimensionality-reduction techniques, massive online dictionary learning assigns non-negative continuous loadings to each voxel designating its relative weight on each region. Voxels with a loading value equal to 0 on any given region do not belong in a given region. Finally, to identify the DiFuMo regions in the LPFC, we recover those that have at least 50% of their volume fall within the LPFC mask described earlier. See the next section on Representing heterogeneous data in a single framework with NeuroLang for more details. Note that we do not mask out region voxels outside the LPFC mask—we either include or exclude entire DiFuMo regions without breaking continuity. This means that some functional regions can include voxels outside the LPFC mask. The reason for this crossover is that functionally defined regions seldom conform to anatomical landmarks in the brain. Comprehensive details on the DiFuMo atlas can be found in the original study by *Dadi et al., 2020*.

## Representing heterogeneous data in a single framework with NeuroLang

The goal behind developing NeuroLang is to create a universal language that reduces the likelihood of miscommunication within the cognitive neuroscience community by enabling databases, hypotheses, and questions to be defined in a formal, shareable, and reproducible manner. This is believed to be a critical step toward advancing the field of cognitive neuroscience (*Poldrack and Yarkoni, 2016*).

In this study, we represent data types from heterogeneous sources, such as peak coordinates, topic models, anatomical masks, and brain atlases in a single framework. More precisely, these data and the relationships among them can be represented as facts and rules using declarative logic-based statements. The user only has to specify what is to be found rather than how to find it. Facts and rules in NeuroLang are tuple sets or tables structured in rows. Each row is a sequence of $k$ elements representing a piece of data, such as the MNI coordinates of a reported peak in a study, and can be implicitly assigned a probability that quantifies the level of uncertainty in this data. Fact tables represent explicit information present in the data, while rule tables represent inferred relationships among different data elements. The goal is to declare these tables as predicates in a probabilistic logic program that solves complex queries on them. For a survey on probabilistic databases and probabilistic programming, the reader is referred to *Van den Broeck and Suciu, 2015*.

To concretely showcase how we represent data in NeuroLang, we start with the Neurosynth dataset. The database includes studies that report peak activations coordinates in standard space (*Figure 6A*). In NeuroLang, we represent these peaks in a fact table called `PeakReported`. This table contains a row (x, y, z, study) for each peak with coordinates (x, y, z) that has been reported active by a `study`. Also, the studies themselves are represented in a fact table called `Study` that contains a row for each study containing a single element, (study), representing its PubMed identifier. Similarly to Neurosynth, we assume each study within the database to be an *independent equiprobable sample* of neuroscientific knowledge (*Yarkoni et al., 2011*). This assumption is represented by another fact table we call `SelectedStudy`, which simply assigns a uniform probability ($1/N$, $N = 14,370$) for each study to be selected in any possible world of events. In other words, this assumption allows the studies to have equal weights in the meta-analysis (*Yarkoni et al., 2011*; *Iovene and Wassermann, 2020*).

Further, the spatial uncertainty surrounding the reported location of each peak in a given study can be represented in a rule table named `VoxelReported`. In this rule table, a multilevel kernel density analysis (MKDA) (*Wager et al., 2007*) assumes each peak's 10mm neighboring voxels to be equivalently reported (*Wager et al., 2007*). Thus, the VoxelReported table contains a row (x, y, z, study) for each voxel at location (x, y, z) and falls within 10mm Euclidean distance from a peak reported by a `study`. As Neurolang is based on Datalog (*Abiteboul et al., 1995*), a declarative logic programming

language designed to solve queries on large databases, the program that computes VoxelReported is written as follows:

```
VoxelReported(x, y, z, study):-
  GreyMatterVoxel(x, y, z) & PeakReported(x2, y2, z2, study)
  & distance = EUCLIDEAN(x, y, z, x2, y2, z2) & distance < 10

ans(x, y, z, study):- VoxelReported(x, y, z, study)
```

The answer states that "a voxel at location (x, y, z) in grey matter is considered active in a study if it is situated within 10mm radius from a peak reported by study at location (x2, y2, z2)". GreyMatterVoxel is also a fact table representing a grey matter mask in MNI space. This table contains a row (x, y, z) for each brain voxel having at least 25% chance of being found in grey matter. The distance variable is estimated using the built-in function EUCLIDEAN, which computes the distance between two locations in standard space.

The Neurosynth dataset also includes topics that have been derived using latent Dirichlet allocation applied to the abstracts of the articles (*Poldrack and Yarkoni, 2016*). Each study in the database has a loading value on each topic, which can be considered as a proxy to the probability that a topic is present in a study. This weighted topic-study association is represented in a probabilistic fact table named TopicInStudy. This table contains a row (topic, study) for each topic present in a study, and the study has a non-zero loading on the topic. The reason for calling this table probabilistic is that the study-on-topic loading is implicitly embedded as a measure of uncertainty in the presence of a topic in a given study.

Anatomical masks and functional atlases can also be represented in NeuroLang. For instance, we represent the LPFC mask described previously in a fact table called LPFCVoxel. This table contains a row (x, y, z) for each voxel belonging to the LPFC mask. Moreover, we represent the 1024 functional regions from DiFuMo in a fact table called RegionVoxel, which contains a row (r, x, y, z, w) for each voxel at location (x,y,z) in MNI space having a non-zero weight w on a DiFuMo region r. Similar to the case of topic-study association, the voxel-on-region weight can be used as a measure of uncertainty in a voxel belonging to a region. This is achieved by first scaling the weight of every voxel in each region to the maximum weight in that region. In this sense, the voxel with the maximum loading will have a probability of 1 of belonging to a given region. Finally, regions belonging to the LPFC can be represented in a rule table called LPFCRegion. This table contains a row (r), where r is a brain region that have at least 50% of its volume overlapping with the LPFC mask. The following NeuroLang program, written in Datalog syntax, infers this table:

```
RegionVolume(r, count(x, y, z)):- RegionVoxel(r, x, y, z, w)

VolumeOfOverlapWithMask(r, count(x, y, z)):-
  RegionVoxel(r, x, y, z, w) & LPFCVoxel(x, y, z)

LPFCRegion(r):-
  RegionVolume(r, v0) & VolumeOfOverlapWithMask(r, v) & (v/v0 >0.5)

ans(r):- LPFCRegion(r)
```

We start by declaring the predicates to be used in solving the query. These predicates are the RegionVolume and VolumeOfOverlapWithMask. These encode the total volume and the volume of overlap with the LPFC mask of each brain region r, respectively. The LPFCRegion is the final answer of this program and states that: "A brain region r belongs to the LPFC if its volume of overlap, v, with an LPFC mask makes up more than 50% of its total volume, v0". Volume variables v and v0 are estimated by the built-in function count(x, y, z), which simply counts the number of voxels in a brain region.

## Estimating the meta-analytic connectivity matrix with NeuroLang

To infer a whole-brain meta-analytic connectivity profile for each LPFC region, we query the database on the probability that a brain region is reported active given the presence or absence of activation in a LPFC region. For this purpose, we write a NeuroLang program that first projects the voxels reported active in each study onto the 1024 functional regions to determine which ones are reported by the study (step 2 in *Figure 6B*). In this context, the program regards the reporting of a brain region by a study as a probabilistic event rather than a deterministic one. That is, if a voxel reported active has a weight w on a functional region r, then the region is assigned a probability w of being reported by the study. If multiple voxels with distinct weights are reported active within a region, then the maximum weight is considered as the probability for the region to be reported by the study. Second, we infer the probabilities of observing activation in every brain region given the presence, and subsequently given the absence, of activation in each LPFC region. We then compute the logarithm of the odds ratio, the difference of the logits of the probabilities. This yields a vector for each LPFC region whose elements represent the amount of evidence for pairwise meta-analytic connectivity with every brain region. A positive LOR indicates more evidence for coactivation, a negative LOR implies more evidence independent activation, and a LOR equal to 0 implies that evidence is inconclusive for either hypotheses. The program that infers the meta-analytic connectivity matrix is as follows:

```
RegionMaxWeight(r, max(w)):- RegionVoxel(r, x, y, z, w)

RegionVoxelNormalizedWeight(r, x, y, z):: w/W:-
  RegionVoxel(r, x, y, z, w) & RegionMaxWeight(r, W)

LPFCRegionActive(r, study):-
  RegionVoxelNormalizedWeight(r, x, y, z)
  & VoxelReported(x, y, z, study)
  & LPFCRegion(r)

LPFCRegionNotActive(r, study):-
  LPFCRegion(r)
  & ~LPFCRegionActive(r, study)
  & Study(study)

BrainRegionActive(r, study):-
  VoxelReported(x, y, z, study)
  & RegionVoxelNormalizedWeight(r, x, y, z)

ProbabilityOfCoactivation(r, r2, PROB):-
  BrainRegionActive(r, study) //
(LPFCRegionActive(r2, study)
& SelectedStudy(study))

ProbabilityOfNoCoactivation(r, r2, PROB):-
  BrainRegionActive(r, study) //
  (LPFCRegionNotActive(r2, study)
  & SelectedStudy(study))

MetaAnalyticConnectivityMatrix(r2, r, LOR):-
  ProbabilityOfCoactivation(r, r2, p1)
  & ProbabilityOfNoCoactivation(r, r2, p0)
  & LOR = log10((p1/(1-p1))/(p0/(1-p0)))

ans(r2, r, LOR):- MetaAnalyticConnectivityMatrix(r2, r, LOR)
```

In order to solve the query, we first need to declare the predicates. First, we get the maximum weight of each DiFuMo brain region in RegionMaxWeight using the built-in function max(w). This will be used to declare a probabilistic table RegionVoxelNormalizedWeight, which implicitly incorporates the normalized weight w/W of each voxel (x, y, z) in each DiFuMo-1024 region r. The normalized weight is represented by the (:: w/W). Second, we define the probabilistic tables LPFCRegionActive and BrainRegionActive, wherein each row carries a probability that a brain or LPFC region is reported active by a study. Similarly, we define LPFCRegionNotActive, a probabilistic table wherein each row carries the probability that a study does not report activation in a LPFC region. We then infer ProbabilityOfCoactivation, which encodes the probability (PROB) of activation being reported in brain region r given (//) that activation is also reported in LPFC region r2. Likewise, we infer ProbabilityOfNoCoactivation, which encodes the probability of activation being reported in a brain region given that activation is not reported in a LPFC region. The SelectedStudy table sets the program to assign an equal weight ($1/N$, $N = 14,371$) to all the studies in the meta-analysis. Finally, the MetaAnalyticConnectivityMatrix rule table is inferred by computing the LOR of the two events (i.e. coactivation and no coactivation) as a measure of meta-analytic connectivity between each LPFC region and every brain region.

## Diffusion map embedding using the BrainSpace toolbox

To recover a low-dimensional embedding of the meta-analytic connectivity matrix, we choose to apply diffusion embedding (*Coifman et al., 2005*), an unsupervised nonlinear dimensionality reduction method. The low-dimensional embeddings represent the axes of variation of coactivation-based connectivity patterns in the LPFC, and can be recovered with two steps. First, we estimate the similarity between LPFC regions in terms of their coactivation patterns. Here, we quantify the similarity between each pair of LPFC regions using the eta-squared coefficient following *Haak et al., 2018*, yielding a square affinity matrix (step 4 in *Figure 6B*). The eta-squared coefficient represents the fraction of the variance in one meta-analytic connectivity profile that is accounted for by the variance in another and ranges from 0 (totally dissimilar) to 1 (perfectly similar). Diffusion embedding then represents this similarity structure as an arrangement of regions in an embedding space spanned by 20 components known as 'gradients'. Gradients are conceptually similar to the components of principal components analysis and represent unidimensional axes, each explaining a fraction of the variance in a given feature (*Margulies et al., 2016*), in our case, meta-analytic connectivity. In each gradient, regions that have very similar meta-analytic connectivity patterns occupy nearby zones, while regions with dissimilar patterns are situated further apart. The first or principal gradient is the most informative component as it captures the dominant axis of variation of meta-analytic connectivity patterns within the LPFC.

## Inferring whole-brain coactivation patterns of quintile bins along the principal gradient using NeuroLang

To be able to infer varying coactivation patterns along the principal gradient in the LPFC, we first create regions-of-interest from successive twenty-percentile gradient bins (i.e. five quintile bins) in the right and left LPFC. Then, we infer each quintile bin's coactivation pattern and characterize the variation of network connectivity along the principal gradient (step 6 in *Figure 6B*). We represent the voxels of the quintile bins in each hemisphere as fact tables LeftBinVoxel and RightBinVoxel for the left and right LPFC, respectively. Each of these tables includes a row (bin, x, y, z) for each voxel at location (x, y, z) in MNI space and belonging to a quintile bin. Moreover, we declare another fact table Bin whose rows contain only the labels of the quintile bins (i.e. bin1 to bin5).

We write a NeuroLang program that infers the conditional probability of a brain region to be reported active given activation reported in a bin as well as when given no bin activation. Specifically, the program infers the LOR of these two hypotheses as a measure of evidence for coactivation. A cortical coactivation pattern for each quintile bin is then constructed by recovering the brain regions that exhibit at least threefold the evidence (or LOR >0.5) of being reported active when given activation in a bin relative to no bin activation. The NeuroLang program that infers coactivation patterns of quintile bins in the left LPFC is as follows:

```
LeftBinActive(bin, study):-
  LeftBinVoxel(bin, x, y, z)
```

```
                              & PeakReported(x2, y2, z2, study)
                              & distance == EUCLIDEAN(x, y, z, x2, y2, z2)
                              & distance < 3

                        LeftBinNotActive(bin, study):-
                          Study(study)
                        & Bin(bin)
                        & ~LeftBinActive(bin, study)

                        BrainRegionActive(r, study):-
                          VoxelReported(x, y, z, study)
                        & RegionVoxelNormalizedWeight(r, x, y, z)

                        ProbabilityOfCoactivation(r, bin, PROB):-
                          BrainRegionActive(r, study) //
                        (LeftBinActive(bin, study)
                        & SelectedStudy(study))

                        ProbabilityOfNoCoactivation(r, bin, PROB):-
                          BrainRegionActive(r, study) //
                        (LeftBinNotActive(bin, study)
                          & SelectedStudy(study))

                        CoactivationPattern(bin, r):-
                          ProbabilityOfCoactivation(r, bin, p1)
                        & ProbabilityOfNoCoactivation(r, bin, p0)
                        & LOR == log10((p1/(1 p1))/(p0/(1 p0)))
                        & LOR > 0.5

                        ans(bin, r):- CoactivationPattern(bin, r)
```

   As in the program of the previous section, we first declare the predicates that will be used to find the answer to our query. We set the program to consider activity in a quintile bin to be reported by a study if at least one peak activation is reported within the bin or within its near vicinity (distance <3). The program stores the results in the rule table LeftBinActive, which includes a row (bin, study) for each bin in which activation is reported by a study. The program then derives the studies that do not report activity within each bin using the negation operator and stores them in another rule table Left-BinNotActivate. This rule table includes a row for each bin wherein no activation has been reported by a study. Similarly as the program in the previous section, the program considers activation reporting in individual brain regions as a probabilistic rather than deterministic event depending on the location of active voxels within each region, and stores the results in the rule table BrainRegionActive. The program then infers the conditional probabilities of the two hypotheses and stores them in the rule tables ProbabilityOfCoactivation and ProbabilityOfNoCoactivation. Finally, the answer to our query CoactivationPattern is derived by estimating the LOR as a measure of evidence in favor of coactivation between each brain region r and each quintile bin and thresholding it at LOR >0.5. Below is a similar program that infers coactivation patterns of quintile bins in the right LPFC:

```
                        RightBinActive(bin, study):-
                          RightBinVoxel(bin, x, y, z)
                          & PeakReported(x2, y2, z2, study)
                          & distance == EUCLIDEAN(x, y, z, x2, y2, z2)
                          & distance < 3

                        RightBinNotActive(bin, study):-
                          Study(study)
                          & Bin(bin)
```

```
      & ~RightBinActive(bin, study)

BrainRegionActive(r, study):-
  VoxelReported(x, y, z, study)
  & RegionVoxelNormalizedWeight(r, x, y, z)

ProbabilityOfCoactivation(r, bin, PROB):-
  BrainRegionActive(r, study) //
  (RightBinActive(bin, study)
  & SelectedStudy(study))
ProbabilityOfNoCoactivation(r, bin, PROB):-
   BrainRegionActive(r, study) //
   (RightBinNotActive(bin, study)
    & SelectedStudy(study))

CoactivationPattern(r, bin, PROB):-
  ProbabilityOfCoactivation(r, bin, p1)
  & ProbabilityOfNoCoactivation(r, bin, p0)
  & LOR == log10((p1/(1 p1))/(p0/(1 p0)))
  & LOR > 0.5

ans(bin, r):- CoactivationPattern(bin, r)
```

## Inferring specific structure-function associations using NeuroLang segregation queries

We infer specific structure-function associations by estimating the extent to which a spatially-localized activation along the principal gradient in the LPFC predicts a Neurosynth topic's presence in a study. For this purpose, we write NeuroLang programs that include what we call 'segregation queries'. Segregation queries infer the probability that a topic is present in a study given spatially constrained activation within a range of quintile bins and the simultaneous absence of activation outside this range within the same hemisphere. Concurrently, a segregation query infers the probability of the opposite event: a topic is present given no activation within the range of quintile bins or there exists activation outside the range. The LOR of these two hypotheses gives us a measure of evidence in favor of association between a topic and patterns of activity along the principal gradient. The NeuroLang program that infers specific structure-function associations in the left LPFC using segregation queries is as follows:

```
LeftBinActive(bin, study):-
  LeftBinVoxel(bin, x, y, z)
  & PeakReported(x2, y2, z2, study)
  & distance == EUCLIDEAN(x, y, z, x2, y2, z2)
  & distance < 3

SegregationRule(bin1, bin2, study):-
  LeftBinActive(bin1, study)
  & LeftBinActive(bin2, study)
  & (bin2 ≥ bin1)
  & ~exists(bin3;
    Bin(bin3) & (bin3<bin1 | bin3>bin2)
    & Study(study) & LeftBinActive(bin3, study))

NoSegregationRule(bin1, bin2, study):-
  Study(study)
  & Bin(bin1) & Bin(bin2)
  & ~SegregationRule(bin1, bin2, study)
```

```
TopicPresentGivenSegregationRule(topic, bin1, bin2, PROB):-
  TopicInStudy(topic, study) //
  (SegregationRule(bin1, bin2, study)
  & SelectedStudy(study))

TopicPresentGivenNoSegregationRule(topic, bin1, bin2, PROB) :-
  TopicInStudy(topic, study) //
  (NoSegregationRule(bin1, bin2, study)
   & SelectedStudy(study))
LeftBinActive(bin, study):-
  LeftBinVoxel(bin, x, y, z)
  & PeakReported(x2, y2, z2, study)
  & distance == EUCLIDEAN(x, y, z, x2, y2, z2)
  & distance < 3

SegregationRule(bin1, bin2, study):-
  LeftBinActive(bin1, study)
  & LeftBinActive(bin2, study)
  & (bin2 ≥ bin1)
  & ~exists(bin3;
    Bin(bin3) & (bin3<bin1 | bin3>bin2)
    & Study(study) & LeftBinActive(bin3, study))

NoSegregationRule(bin1, bin2, study):-
  Study(study)
  & Bin(bin1) & Bin(bin2)
  & ~SegregationRule(bin1, bin2, study)

TopicPresentGivenSegregationRule(topic, bin1, bin2, PROB):-
  TopicInStudy(topic, study) //
  (SegregationRule(bin1, bin2, study)
  & SelectedStudy(study))

TopicPresentGivenNoSegregationRule(topic, bin1, bin2, PROB) :-
  TopicInStudy(topic, study) //
  (NoSegregationRule(bin1, bin2, study)
   & SelectedStudy(study))

TopicAssociationMatrix(topic, bin1, bin2, LOR) :-
  TopicPresentGivenSegregationRule(topic, bin1 , bin2, p1)
  & TopicPresentGivenNoSegregationRule(topic, bin1, bin2, p0)
  & LOR == log10((p1/(1 - p1))/(p0/(1 - p0)))

ans(topic, bin1, bin2, LOR) :- TopicAssociationMatrix(topic, bin1, bin2,
LOR)
```

We first declare the studies that report activations in each quintile bin of the left principal LPFC gradient in a rule table LeftBinActive. Then, we declare a segregation query which first identifies the studies that report coactivation between each pair of quintile bins along the principal gradient, named bin1 and bin2, under the conditions that (bin2 ≥ bin1) and there exists no activation reported in any bin3, such that bin3 <bin1 or bin3 >bin2. That is, activation in any bin that is outside the range [bin1, bin2] should not be present according to the segregation rule, whereas activation within the range between the bins can exist. Here, 'there exists no' is represented by "~exists", which is a combination of the negation operator and the existential quantifier. The results are represented in a SegregationRule rule table, which includes a row (bin1, bin2, study) for bins (bin1, bin2) between

which activation is reported in study that also satisfies the segregation condition. Concurrently, we declare the studies that do not match the conditions of the segregation query, and represent them in the rule table NoSegregationRule. After defining the predicates, the program infers the conditional probability that a topic is present in a study given the presence as well as absence of the segregation condition. The results are represented in the tables TopicPresentGivenSegregationRule and TopicPresentGivenNoSegregationRule. Finally, the answer to our query, represented in the rule table TopicAssociationMatrix, is derived by computing the LOR of the two hypotheses as a measure of evidence in favor of association between each topic and activation in a quintile range [bin1, bin2] along the principal gradient. The NeuroLang program that infers specific topic associations of coactivation patterns within the right LPFC is as follows:

```
RightBinActive(bin, study):-
  RightBinVoxel(bin, x, y, z)
& PeakReported(x2, y2, z2, study)
& distance == EUCLIDEAN(x, y, z, x2, y2, z2)
& distance < 3

SegregationRule(bin1, bin2, study):-
  RightBinActive(bin1, study)
  & RightBinActive(bin2, study)
  & (bin2 ≥ bin1)
  & ~exists(bin3;
   Bin(bin3)
   & (bin3<bin1 | bin3>bin2)
   & Study(study) & RightBinActive(bin3, study))

NoSegregationRule(bin1, bin2, study):-
  Study(study)
  & Bin(bin1)
  & Bin(bin2)
  & ~SegregationRule(bin1, bin2, study)

TopicPresentGivenSegregationRule(topic, bin1, bin2, PROB):-
  TopicInStudy(topic, study) //
  (SegregationRule(bin1, bin2, study)
  & SelectedStudy(study))

TopicPresentGivenNoSegregationRule(topic, bin1, bin2, PROB):-
  TopicInStudy(topic, study) //
   (NoSegregationRule(bin1, bin2, study)
    & SelectedStudy(study))

TopicAssociationMatrix(topic, bin1, bin2, LOR):-
  TopicPresentGivenSegregationRule(topic, bin1, bin2, p1)
  & TopicPresentGivenNoSegregationRule(topic, bin1, bin2, p0)
  & LOR == log10((p1/(1-p1))/(p0/(1-p0)))

ans(topic, bin1, bin2, LOR):- TopicAssociationMatrix(topic, bin1, bin2,
LOR)
```

Finally, we write a program that applies inter-hemispheric segregation queries to infer the probability "that a topic is present given activation in a right lpfc quintile bin and there exists no activation in the entire left lpfc." The program also infers the probability of the opposite event; "a topic is present given activation in a left LPFC quintile bin, and there exists no reported activation in the entire right LPFC." The neurolang program that infers hemisphere-specific topic-bin associations is as follows:

```
LeftBinActive(bin, study):-
  LeftBinVoxel(bin, x, y, z)
  & PeakReported(x2, y2, z2, study)
  & distance == EUCLIDEAN(x, y, z, x2, y2, z2)
  & distance < 3

RightBinActive(bin, study):-
  RightBinVoxel(bin, x, y, z)
  & PeakReported(x2, y2, z2, study)
  & distance == EUCLIDEAN(x, y, z, x2, y2, z2)
  & distance < 3

OnlyLeftBinActive(bin, study):-
  LeftBinActive(bin, study)
  & ~exists(bin2;
    Bin(bin2)
    & Study(study) & RightBinActive(bin2, study))

OnlyRightBinActive(bin, study):-
  RightBinActive(bin, study)
  & ~exists(bin2;
    Bin(bin2)
  & Study(study) & LeftBinActive(bin2, study))

TopicPresentGivenOnlyLeftBinActive(topic, bin, PROB):-
  TopicInStudy(topic, study) //
  (OnlyLeftBinActive(bin, study)
  & SelectedStudy(study))

TopicPresentGivenOnlyRightBinActive(topic, bin, PROB):-
  TopicInStudy(topic, study) //
  (OnlyRightBinActive(bin, study)
   & SelectedStudy(study))

InterHemisphereTopicBinAssociation(topic, bin, LOR):-
  TopicPresentGivenOnlyRightBinActive(topic, bin, p1)
& TopicPresentGivenOnlyLeftBinActive(topic, bin, p2)
& LOR == log10((p1/(1-p1))/(p2/(1-p2)))

ans(topic, bin, LOR):- InterHemisphereTopicBinAssociation(topic, bin,
LOR)
```

In this program, we define the predicates LeftBinActive and RightBinActive that represent the studies reporting activation in each quintile bin of the principal gradient in the left and right LPFC, respectively. Then we declare the inter-hemispheric segregation queries using the negation operator and the existential quantifier, ~exists, and stores the results in OnlyLeftBinActive and OnlyRightBinActive. Subsequently, the program infers the conditional probabilities that a topic is present in a study when given activation either in a left or a right quintile bin. The final answer, InterHemisphereTopicBinAssociation, is derived by computing the LOR of the two hypotheses.

## Data availability statement

All data used in this study is available in open-source databases. Meta-analytic data comes from Neurosynth (*Yarkoni et al., 2011*), and the human data comes from the Individual Brain Charting database (*Pinho et al., 2020*). Code for NeuroLang version 0.1a11 is freely available at https://github.com/NeuroLang/NeuroLang; *Wassermann et al., 2022*. In-depth details on NeuroLang are forund

in *Abdallah et al., 2022* and *Iovene, 2021*. All code was developed based on open-source, publicly available software packages.

## Acknowledgements

This work is funded by the ERC-2017-STG NeuroLang grant. We are grateful to Jonas Renault who worked on optimising NeuroLang's engine and develop its web interface. We also want to thank Raphael Meudec for his help in setting up the single-subject analysis.

## Additional information

### Funding

| Funder | Grant reference number | Author |
|---|---|---|
| European Research Council | 10.3030/757672 | Majd Abdallah Demian Wassermann |

The funders had no role in study design, data collection and interpretation, or the decision to submit the work for publication.

### Author contributions

Majd Abdallah, Conceptualization, Formal analysis, Validation, Investigation, Visualization, Methodology, Writing - original draft, Project administration, Writing - review and editing; Gaston E Zanitti, Software, Investigation, Methodology; Valentin Iovene, Conceptualization, Software, Investigation, Methodology; Demian Wassermann, Conceptualization, Resources, Software, Formal analysis, Supervision, Funding acquisition, Validation, Investigation, Methodology, Project administration, Writing - review and editing

### Author ORCIDs

Majd Abdallah http://orcid.org/0000-0002-8371-1793
Gaston E Zanitti http://orcid.org/0000-0001-5549-9548
Demian Wassermann http://orcid.org/0000-0001-5194-6056

### Ethics

Human subjects: The current study uses brain activation data from the Individual Brain Charting Dataset (IBC). In the original paper of IBC, the authors indicate that they received written consent from the subjects involved in the study. To quote from Pinho et al. Individual Brain Charting, a high-resolution fMRI dataset for cognitive mapping. Sci Data. 2018 : "The experimental procedures were approved by a regional ethical committee for medical protocols in Icircumflex;le-de-France ("Comité; de Protection des Personnes" - no. 14-031) and a committee to ensure compliance with data-protection rules ("Commission Nationale de l'Informatique et des Liberté;s" - DR-2016-033). They were undertaken with the informed written consent of each participant according to the Helsinki declaration and the French public health regulation. Participants were reimbursed on the basis of 80 per MRI acquisition with extra-fees for any additional session."

### Decision letter and Author response

Decision letter https://doi.org/10.7554/eLife.76926.sa1
Author response https://doi.org/10.7554/eLife.76926.sa2

## Additional files

### Supplementary files

• Transparent reporting form

## Data availability

All data and scripts used in this study are openly available to be accessed and freely used by the community. The source code of NeuroLang is freely available on GitHub at https://github.com/NeuroLang/NeuroLang.

The following previously published dataset was used:

| Author(s) | Year | Dataset title | Dataset URL | Database and Identifier |
|---|---|---|---|---|
| Yarkoni T, Poldrack RA, Nichols TE, Essen DCV, Wager TD | 2011 | Neurosynth | https://github.com/neurosynth/neurosynth-data | GitHub, neurosynth-data |

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

## Appendix 1

### The principal LPFC gradient is robust and reproducible across different methodological paradigms

To assess potential biases that can arise from the choice of data and methods, we perform the following three supplementary validation analyses.

#### Bootstrap Analysis via Sub-sampling of the Neurosynth Dataset

To assess the variation of the resultant principal gradient when the choice of studies included in a meta-analysis varies, we create 5000 random sub-samples of the Neurosynth dataset and re-run the meta-analysis and gradient mapping on each one. The results are summarized in *Figure 1—figure supplement 1* and *Figure 1—figure supplement 2*. *Figure 1—figure supplement 1* shows that most LPFC regions in both hemispheres are assigned to the same quintile bin along the principal gradient in the majority of sub-samples. The results shown in *Figure 1—figure supplement 2* show that the percentage of variance explained by diffusion embedding components does not vary significantly across the sub-samples.

#### Accounting for Spatial Auto-correlation within the LPFC

To account for the possibility of bias due to spatial auto-correlation among nearby LPFC regions, we re-perform the gradient-mapping using longer coactivation distances between regions. The coactivation distance is the Euclidean distance between the centers of mass of each region. The results are shown in *Figure 1—figure supplement 3* and reveal that although the shape of principal LPFC gradient at different coactivation distances (20mm, 40mm, and 60mm) varies, its general rostrocaudal profile is preserved. This suggests that spatial auto-correlation among nearby regions does not induce systematic bias to the overall shape of the principal gradient.

#### Single-Subject Meta-Analysis and Gradient Mapping

To show that the principal LPFC gradient is not a result of spatial smoothing arising from peak location uncertainty across different studies, we re-infer the gradient at the single subject level (*Figure 1—figure supplement 4* and *Figure 1—figure supplement 5*) using data from the Individual Brain Charting (IBC) dataset (*Pinho et al., 2020*). The IBC subjects underwent an extensive set of tasks ranging from simple motor tasks to long-term memory and social cognition tasks. The single-subject principal LPFC gradients are shown in *Figure 1—figure supplement 4*. These gradients are seemingly different from the literature-inferred principal gradient, exhibiting more granularity and abrupt transitions than the latter. However, *Figure 1—figure supplement 5* shows a moderate-to-strong correlation between most single-subject gradients and the literature gradient in both LPFC hemispheres. This indicates that the gross spatial layout of the principal LPFC gradient is spatially determined at the single subject level. In contrast, the spatial correlation of single-subject gradients with a randomized literature-driven gradient is significantly lower (*Figure 1—figure supplement 5*). The latter gradient is estimated from a randomized version of the Neurosynth database, where peak coordinates are shuffled 1000 times between studies. More details on the single-subject meta-analysis are found below.

### The Individual Brain Charting Dataset

We assess the reproducibility of the principal LPFC gradient at the subject-level by re-running the meta-analysis and the gradient mapping on activation data from the Individual Brain Charting (IBC) dataset (*Pinho et al., 2018*; *Pinho et al., 2020*). The IBC dataset is a free and open dataset devoted to providing a deep phenotyping characterization of brain systems within the scope of cognitive neuroscience. In the IBC project, 12 subjects are scanned extensively, and a huge number of imaging contrasts (both anatomical and functional) are extracted in order to create a sharp definition of the role of each brain region. The dataset contains dozens of tasks, addressing both low- and high- level cognitive functions, including motor tasks, working memory, language tasks, relational tasks, social tasks, mental time travel, reward, theory-of-mind, pain, numerosity, self-reference effect and speech recognition (*Pinho et al., 2018*; *Pinho et al., 2020*). This extensive set of tasks yielded a total of 750 contrasts for each individual subject. These contrasts are publicly available on NeuroVault at https://neurovault.org/collections/6618/.

## Peak Extraction from the IBC dataset

After downloading the subject-level contrasts and before re-running the coordinate-based meta-analysis, we need to extract the peak activations locations in MNI space. For this purpose, we perform cluster-extent-based thresholding that controls the family-wise error rate (FWER) (**Woo et al., 2014**). In this context, we use a primary threshold $\alpha < 0.001$, a fixed cluster volume threshold equal to 5 $mm^3$, minimum distance between peaks of 10mm, and a spherical smoothing kernel with $FWHM = 4mm^3$ for all subjects. We use nilearn (version 0.9.1) python package to perform the peak extraction.

## Single-Subject Meta-Analysis and Gradient Mapping of the LPFC

The single-subject meta-analysis and gradient mapping of the LPFC follow the same steps and use the same queries as the literature-based analyses (**Figure 6**). This includes applying a 10mm spherical smoothing kernel around each peak, projecting the binary activation maps onto the 1024 DiFuMo atlas, inferring a meta-analytic connectivity matrix for each subject, computing a similarity matrix, and finally estimating the gradients using diffusion embedding. However, the main difference is that instead of using studies as in conventional meta-analysis, the unit of analysis here is the individual contrast. For each subject, the $\approx 750$ contrasts are treated as separate studies carrying the same weight in a meta-analysis. While this assumption might be imperfect, it serves to simplify the analysis by assuming that each contrast represents an *independent equiprobable sample* of neuroscientific knowledge. We only focus on the principal LPFC gradient of each subject for the current study. Future investigations should embark on more extensive attempts to map the brain's organizing principles in the single subject.

