## [Editor Report]

A meta-analysis of over 14,000 fMRI studies revealed a principle rostral-caudal gradient in the lateral prefrontal cortex. This gradient reflected an internal/external axis, which helps to organize the LPFC's involvement in widespread processes from affect, to memory, to control, and action. This is an important contribution to the literature, particularly as a meta-analytic approach has not been applied to this axis of organization and can complement the limitations of single studies. The paper is strengthened by the authors addressing a potential bias in the analysis and drawing a clearer relationship to functional networks.

---

## [Decision Letter]

**Decision letter after peer review:**

Thank you for submitting your article "Functional gradients in the human lateral prefrontal cortex revealed by a comprehensive coordinate-based meta-analysis" for consideration by *eLife*. Your article has been reviewed by 2 peer reviewers, and the evaluation has been overseen by David Badre as Reviewing Editor and Michael Frank as the Senior Editor. The reviewers have opted to remain anonymous.

The reviewers have discussed their reviews with one another and the Reviewing Editor, and the Reviewing Editor has drafted this to help you prepare a revised submission. The reviewers were in consensus that this paper makes an important contribution to this literature, and with revision, may be suitable for publication.

Essential revisions:

The essential revisions focused on two areas. First, they felt that revisions were needed to rule out any bias in the analysis (for example, a spatial contiguity bias) and to show specificity. Second, they thought that a clearer relationship to functional networks, macroscale gradients, and cognitive terms could be established. The specific essential revisions from each reviewer to address these points are below.

1) I struggled a bit with what to make of Figure 3. The authors suggest "This analysis reveals a structured ordering of network connectivity profiles along the rostrocaudal LPFC gradient in both hemispheres, from a pattern mostly dominated by networks involved in external processing to a pattern dominated by networks involved in internally-oriented cognition." While I am sympathetic to this position, I did not see it in the data as presented. To my eyes, it seemed that the co-activations of bins 1 and 2 were more widespread among brain networks, whereas more anterior bins produce activations that overlapped with more circumscribed networks. I feel as though Figure 2 did a somewhat better job of supporting the authors' position. In both cases, however, some easily digestible quantification would seem to help. Although the authors attempt to denote overlap using color transparency, I had a difficult time parsing that element. I think some bar graphs would be helpful in this regard. Moreover, although the authors are using the colors typically associated with Yeo17, I find that the color palette is hard to map onto a gradient. Since the authors have ordered the networks along a presumed external to internal dimension, it might make sense to use a color palette that similarly continuously ordered the networks. That might help to make one end of the color palette more obviously external and another end more obviously internal.

2) Similar to the above, I felt as though a more concrete means to quantify the patterns observed in Figure 4 would be helpful. The authors conclude that the data show "a systematic shift in topic associations from external processing at the caudal end to more abstract cognitive, affective and memory-related topics at the rostral end of the principal gradient. Between these extremities, we observe domain-general executive functions and topics related to language and semantic processing." Perhaps there is an objective way to weight each of the terms with the descriptions of external processing, cognitive, affective, memory-related, domain-general executive function, language, and semantic. Or perhaps a condensed set of those terms. Anything that does not rely on a subjective eyeball assessment of the terms would be welcome. Moreover, I found the term "domain-general executive function" particularly loaded. What would objectively constitute domain-general vs domain-specific?

3) The meta-analytic techniques utilized here strike me as a marked innovation. However, at times I was left wondering how the techniques utilized would compare to something that is a bit more bog-standard. For example, when the authors use "segregation queries", they make the argument that these allow more specific associations than would result from standard queries. Perhaps as a supplemental figure, it would be useful to see what a non-segregated query looks like. If it is indeed very non-specific, this helps to make the case for why more selective tools are necessary.

4) Given the coarseness of coordinate-based meta-analyses and derivations thereof, I wondered whether the methods are biased to show continuous-looking gradients. That is, there is spatial smearing at multiple levels. First, in terms of atlas. Next, in terms of MKDA. I wonder the extent to which that smearing hides potential sharp boundaries. For example, a marked aspect of the Yeo17 is that multiple networks spanning different functions can be found in close spatial proximity (c.f. dorsal-caudal PFC containing Dorsal Attention, Control B, Default B, and Control A). If simulated data had such granularity, would it be missed by these methods? If so, I think it is an important caveat to add to the limitations section: the methods may make the organization of the LPFC appear more spatially ordered than really is the case.

5) In the introduction the authors mention that networks such as SN and FPN sit upon global connectivity gradients, notably the widely recognized principal gradient that runs from sensory to heteromodal regions. One broader question of the current work may be how far the LPFC gradients are mere reflections of global gradients such as the first principal gradient but also the third functional gradient (which differentiates multiple demand networks that subsume FPN) described by Margulies and colleagues, or whether these region-specific gradients are specific cases. In other words, the authors are recommended to also perform a systematic spatial correlation analysis between the LPFC gradients derived from meta-analysis and these previously described rs-fmri gradients in the LPFC mask and to more broadly discuss the interplay of local and more global organizational axes.

6) Meta-analysis is arguably the best technique to synthesize findings across individual studies, but doesn't it also amplify spatial uncertainty in functional localization and thus naturally favour more gradient-like arrangements? Several prior studies have suggested that higher association networks, including some of those that naturally make up LPFC, may often have rather fine-grained (e.g. inter-digitated) spatial patterns, which may be washed out when averaging different datasets. In addition to discussing this issue more extensively, it may make sense to also explore whether subject-specific functional analysis (e.g. based on HCP or other datasets with extensive task fmri per participant) could be a meaningful complement to the current analysis, as it would help to solidify whether the observed functional gradients are also as spatially determined when studying individual subject level functional topographies.

7) Related to the above point is the clear spatial anchoring of the gradients. Are similar gradients also observed when e.g. focusing only on meta-analytical co-activation of longer-range connections? Systematically varying the connectivity distance thresholds could be something quite meaningful here, as it may reduce potential spatial autocorrelation in observed networks.

8) The analysis on page 6 had a slight suboptimal flavour, as network participation was equated with abstraction. In my view, the subsequent definition of abstraction via cognitive terms is better, so I would possibly rephrase the section and avoid reverse inference (e.g. more dmn, more abstraction).

9) The inter-hemispheric asymmetry analysis is interesting and novel. One question that one may want to ask is how far the underlying image processing of the included studies, as well as the meta-analytical synthesis, accounted for potential asymmetries in the brain. In my understanding, a requirement for asymmetry analysis is the mapping of imaging data to a hemisphere unbiased (ie symmetric) template. Is such a step ensured in the included studies and in the meta-analysis inference itself? For example, when carrying out analyses in surface space (results are shown on surfaces) the surface vertices need to ideally be matched across hemispheres to avoid mismatch.

10) Wouldn't an additional bootstrap across the studies that were included be useful in order to further model uncertainty across studies (e.g. with respect to the variance explained and gradient shape for Figure 1, topic associations in Figure 4, and assymetries in Figure 5).

---

## [Author Response]

Essential revisions:The essential revisions focused on two areas. First, they felt that revisions were needed to rule out any bias in the analysis (for example, a spatial contiguity bias) and to show specificity. Second, they thought that a clearer relationship to functional networks, macroscale gradients, and cognitive terms could be established. The specific essential revisions from each reviewer to address these points are below.(1) I struggled a bit with what to make of Figure 3. The authors suggest "This analysis reveals a structured ordering of network connectivity profiles along the rostrocaudal LPFC gradient in both hemispheres, from a pattern mostly dominated by networks involved in external processing to a pattern dominated by networks involved in internally-oriented cognition." While I am sympathetic to this position, I did not see it in the data as presented. To my eyes, it seemed that the co-activations of bins 1 and 2 were more widespread among brain networks, whereas more anterior bins produce activations that overlapped with more circumscribed networks. I feel as though Figure 2 did a somewhat better job of supporting the authors' position. In both cases, however, some easily digestible quantification would seem to help. Although the authors attempt to denote overlap using color transparency, I had a difficult time parsing that element. I think some bar graphs would be helpful in this regard. Moreover, although the authors are using the colors typically associated with Yeo17, I find that the color palette is hard to map onto a gradient. Since the authors have ordered the networks along a presumed external to internal dimension, it might make sense to use a color palette that similarly continuously ordered the networks. That might help to make one end of the color palette more obviously external and another end more obviously internal.

We agree with the reviewer(s) that Figure 3 in its original form may have been misleading and did not clearly convey the results as we discussed them. For this reason, we have refrained form denoting network overlap using color transparency on a brain plot. Instead, we have added bar plots of the number of regions from each canonical brain network that falls within its coactivation pattern of each quintile bin along the principal gradient. However, we have opted to preserve the original color palette of Yeo-17 atlas to be consistent with its wide use in the neuroimaging literature. Figure 3 in its current revised version clearly shows that the dorsal attention and sensorimotor networks dominantly overlap with the coactivation patterns of caudal quintile bins of the LPFC more than with those of more rostral bins. As we move anteriorly, the overlap with these networks fades away and the number of default mode networks regions steadily increases, peaking at the rostral quintile bins 4 and 5. We believe these data support a unimodal-totransmodal mode of network connectivity along the rostrocaudal LPFC gradient. We have modified the caption of Figure 3 to further clarify the results it presents, and now we write:

“(B) Bar plots showing the number of regions from each network that is covered by the coactivation pattern of each quintile bin. The data shown here suggests that the dorsal attention (green) and sensorimotor network (blue) coactivate with the caudal bins (i.e. bins 1 and 2) more than with more rostral ones. On the other hand, the default mode network coactivates more with the rostral bins (i.e. bins 4 and 5) than with caudal ones."

(2) Similar to the above, I felt as though a more concrete means to quantify the patterns observed in Figure 4 would be helpful. The authors conclude that the data show "a systematic shift in topic associations from external processing at the caudal end to more abstract cognitive, affective and memory-related topics at the rostral end of the principal gradient. Between these extremities, we observe domain-general executive functions and topics related to language and semantic processing." Perhaps there is an objective way to weight each of the terms with the descriptions of external processing, cognitive, affective, memory-related, domain-general executive function, language, and semantic. Or perhaps a condensed set of those terms. Anything that does not rely on a subjective eyeball assessment of the terms would be welcome. Moreover, I found the term "domain-general executive function" particularly loaded. What would objectively constitute domain-general vs domain-specific?

We strongly agree with the reviewer that an objective way to link the topics to their ascribed descriptions would be meaningful and interesting for the community. However, we believe that such a step requires a standalone comprehensive meta-analysis. The main reason is that creating a model to probabilistically weigh the topics with respect to the descriptions is not a trivial procedure. The topic terms used herein are themselves data-driven aggregations of singular terms that frequently co-occur in the abstracts of articles. For example, the topic “Eye Movements” is a set of terms, such as “eye”, “gaze”, “saccades”, “movements”, and “fixation”, among many others. To derive a more general level of topic aggregations, a hierarchical topic modeling analysis is needed. This will enable us to estimate a hierarchical structure combining simple terms into topics that are further grouped in broader topics (i.e. descriptions). Although such an analysis helps in ruling out subjective assessments of topic-bin associations, its potential technical complexity may distract the readers from the main message of the study. Nevertheless, it would be very interesting for us to pursue such a research question in the future. For now, we have added a table to Methods section on pages 14 and 15, showing the top 5 single terms belonging to each topic. The goal of the table is to show that the descriptions ascribed to the topics are mainly inspired by the top five loading terms on each topic. The complete list of terms will be provided along the source code and data at https://github.com/majdabd/lpfc_gradients-metaanalysis.

Further, we agree that the term “domain-general executive function” may have been over-loaded. So we have dropped the explicit use of “domain-general” when referring to executive functions. However, we have adopted the nomenclature used in the literature, such as in Nee 2021 [1] and Margulies et al., 2016 [4], with regards to the use of “domain-general” and “domain-specific” when describing mental functions. To explain this further, what we call “domain-specific” functions are those that involve components of either external/present-oriented or internal/temporally-remote processing, whereas ‘domain-general” functions are those that involve components related to both. We now clarify the difference on page 6 lines 105 to 107 near the first use of the terms “domain-general” and “domain-specific”:

“Here, “domain-specific" denotes involvement in either internal/present-oriented or external/future-oriented processing, whereas "domain-general" is used to indicate involvement in both types of processes.”

(3) The meta-analytic techniques utilized here strike me as a marked innovation. However, at times I was left wondering how the techniques utilized would compare to something that is a bit more bog-standard. For example, when the authors use "segregation queries", they make the argument that these allow more specific associations than would result from standard queries. Perhaps as a supplemental figure, it would be useful to see what a non-segregated query looks like. If it is indeed very non-specific, this helps to make the case for why more selective tools are necessary.

We are happy that the reviewers are impressed by the meta-analytic techniques used in this study. The main reason behind adopting NeuroLang for this type of studies is its expressive power, including the ability to formally express what we call “segregation queries”. Qualitatively, we believe that in the case of the lateral prefrontal cortex, imposing segregation is essential to infer relatively specific functional associations among sets of regions that often co-activate across several tasks [2,3]. The coactivation among LPFC regions may mask out the unique domains of functions that each is involved in. Thus, without imposing segregation, it is hard to know whether an inferred functional association is more specific to an area within the LPFC than to others. However**,** we agree that it may be hard for the readers to discern whether “segregation queries” outperform “non-segregation queries” in such experiments without proper quantitative assessment. For this purpose, we have repeated the topic-gradient association experiment without using segregation queries. The results of this experiment are shown in Figure 4—figure supplement 1. Interestingly, the results of this analysis show an ordering of topics along a concrete-to-abstract axis, but the individual associations are clearly more distributed across bins, indicating less specificity than the segregation-based analysis. We write on page 10 line 235-238:

“Results of the non-segregation analysis are shown in Figure 4—figure Supplement 1 and show a similar ordering of topics along a roughly concrete-to-abstract axis. However, the individual associations are rather distributed across bins, suggesting no specificity of structure-function associations in the LPFC.”

We also write in the discussion on page 12 lines 343-346:

“As shown in Figure 4—figure Supplement 1, topic associations are either equal to zero for all bins or cover the entire LPFC gradient, suggesting no specificity in structure-function associations. Thus, our results stress the importance of using segregation queries to infer relatively specific associations often masked in conventional functional decoding analyses.”

(4) Given the coarseness of coordinate-based meta-analyses and derivations thereof, I wondered whether the methods are biased to show continuous-looking gradients. That is, there is spatial smearing at multiple levels. First, in terms of atlas. Next, in terms of MKDA. I wonder the extent to which that smearing hides potential sharp boundaries. For example, a marked aspect of the Yeo17 is that multiple networks spanning different functions can be found in close spatial proximity (c.f. dorsal-caudal PFC containing Dorsal Attention, Control B, Default B, and Control A). If simulated data had such granularity, would it be missed by these methods? If so, I think it is an important caveat to add to the limitations section: the methods may make the organization of the LPFC appear more spatially ordered than really is the case.

The issue of smoothness of the gradients is typical to this kind of studies. Indeed, smearing from spatial smoothing and the sub-sequent use of the DiFuMo-1024 atlas can introduce artificial continuity to the shape of the gradient. We have added a paragraph to the Limitations sub-section of the Discussion to describe the caveats of adopting these methods. These caveats mainly include masking out potentially abrupt boundaries, giving an impression of perfectly smooth gradients. The boundaries can be blurred for instance due to the assumption that all voxels within a DiFuMo-1024 region carry the same gradient weight. However, the actual spatial resolution could be even smaller or even larger than region, respecting canonical network boundaries [4] for instance. Therefore, we do not make any claims about the actual spatial resolution of the principal

LPFC gradient. We write on page 13 lines 389-400 in the limitations section:

“First, although meta-analysis is arguably one of the best techniques to synthesize findings across the literature, it amplifies spatial uncertainty by combining a large number of different datasets and hence might overly smooth the gradients. In addition, the methods adopted here such as the spatial smoothing prior (10 mm radius) and the use of a brain atlas may further exacerbate the problem and make the organization of the LPFC appear more spatially continuous than it actually might be. Indeed, the actual steps along the rostrocaudal axis might be discrete/ areal, separating disparate networks as in clustering-based representations (e.g., Yeo 17 atlas), may approach total continuity, or something in between. Although the supplementary subject-level analysis shows that individual gradients are more granular than the literature-based gradient (Figure 1—figure supplement 4), we do not make any claims about the ground truth spatial resolution of the principal LPFC gradient, as no evidence has been observed to support any of the hypotheses.”

On another note, we believe that gradient-based mapping and clustering techniques complement each other, rather than being standalone alternatives [4,5]. While clustering methods aim to maximize independence of components (e.g. blind source separation), gradient mapping techniques emphasize similarity/dissimilarity among components. Clustering-based parcellations indeed have their applications in neuroimaging, but by assigning each cortical voxel a unique network membership, they prevent a full realization of the organizing principles underlying brain activity and dynamics. Allowing coupling and overlap across regions/networks may better model dynamic functional interactions that are crucial for task-dependent integration [5, 6]. Overall, while we acknowledge the caveats of the methods, we do not make any claims about the ground truth spatial resolution of the gradients.

(5) In the introduction the authors mention that networks such as SN and FPN sit upon global connectivity gradients, notably the widely recognized principal gradient that runs from sensory to heteromodal regions. One broader question of the current work may be how far the LPFC gradients are mere reflections of global gradients such as the first principal gradient but also the third functional gradient (which differentiates multiple demand networks that subsume FPN) described by Margulies and colleagues, or whether these region-specific gradients are specific cases. In other words, the authors are recommended to also perform a systematic spatial correlation analysis between the LPFC gradients derived from meta-analysis and these previously described rs-fmri gradients in the LPFC mask and to more broadly discuss the interplay of local and more global organizational axes.

We have taken this recommendation into consideration and performed a spatial correlation between the principal LPFC gradient and the macroscale intrinsic connectivity gradients 1 and 3. The results of this supplemental analysis are shown in Figure 2—figure supplement 1. We observe a strong positive correlation with the principal macroscale gradient, and a weak negative correlation with the third gradient, for both LPFC hemispheres. We write on page 4-5 lines 150-161:

“One interesting question is whether the principal gradient is a local stream within the macroscale gradients described by Margulies et al., (2016), like the first gradient (separates sensorimotor from default mode regions) or the third gradient (separates task-positive regions from the default mode network), or whether it is a region-specific gradient. So, within an LPFC mask, we compare the spatial layout of principal LPFC gradient (Figure 2—figure Supplement 1) with that of the macroscale resting-state gradients from Margulies et al., (2016), namely gradient 1 and gradient 3. We observe a moderate positive correlation with gradient 1, and a weak negative correlation with gradient 3, in both left and right LPFC. These data reveal that the distribution of activity in the LPFC is dominated by a rostrocaudal processing axis that fits within the macroscale gradient separating sensorimotor systems from higher-order association regions often implicated in abstract mental functions (Margulies et al., 2016; Huntenburg et al., 2018).”

The supplementary results also support the arguments we make in the revised Discussion section entitled “The principal gradient of meta-analytic connectivity in the LPFC echoes the principal gradient of brain-wide intrinsic connectivity” in which we discuss the interplay between the local LPFC gradient and the more global brain organization. In particular, we write on page 11 lines 318-339:

“As seen in Figure 2—figure Supplement 1, the topography of the principal LPFC gradient correlates with the overall layout of the principal brain-wide gradient described in (Margulies et al., 2016), which represents the dominant spatial principle governing resting-state connectivity throughout the entire cerebral cortex. This spatial principle conceptualizes higherorder cognition as emerging from dynamic interactions of large-scale networks, systematically organized along an axis of abstraction that extends from unimodal sensorimotor regions to transmodal default mode regions (Mesulam, 1998; Margulies et al., 2016; Huntenburg et al., 2018). Importantly, it incorporates the seemingly isolated local processing streams across the cortex within a global continuous frame work. In this sense, the spatial location of a brain region is not arbitrary; a regions’s position along the principal gradient is a major determinant of its connectivity profile, its network membership and consequently its functional role. Specifically, it has been found that the longer the spatial distance between a region and the primary cortices, the more distant are its functional connections and the more it is dispositioned to subserve abstract mental functions (Oligschläger et al., 2017). The default mode network occupies the top end of the global intrinsic connectivity gradient and exhibits the greatest geodesic distance from the sensorimotor cortices, allowing it to process highly_-_internalized information abstracted from immediate sensory input (Margulies et al., 2016; Smallwood et al., 2021). In a similar sense, we find that the rostrocaudal gradient in the LPFC captures systematic transitions in large-scale functional networks (Figure 2 and Figure 3), such that caudal zones mainly coactivate with sensorimotor/attention networks, middle zones mainly coactivate with salience/executive control networks, and most rostral zones coactivate more with the default mode network. This result may indeed confer a spectrum of increasing abstraction in the LPFC that follows the transition from unimodal to heteromodal to transmodal regions.”

(6) Meta-analysis is arguably the best technique to synthesize findings across individual studies, but doesn't it also amplify spatial uncertainty in functional localization and thus naturally favour more gradient-like arrangements? Several prior studies have suggested that higher association networks, including some of those that naturally make up LPFC, may often have rather fine-grained (e.g. inter-digitated) spatial patterns, which may be washed out when averaging different datasets. In addition to discussing this issue more extensively, it may make sense to also explore whether subject-specific functional analysis (e.g. based on HCP or other datasets with extensive task fmri per participant) could be a meaningful complement to the current analysis, as it would help to solidify whether the observed functional gradients are also as spatially determined when studying individual subject level functional topographies.

We strongly agree that a single-subject analysis can be meaningful in order to establish the robustness of the gradients beyond aggregated data. For this purpose, we re-performed the same coordinate-based meta-analysis using activation data of 11 subjects from the Individual Brain Charting database IBC [7]. The IBC subjects underwent a large number of tasks that yielded an average of 750 contrasts per subject. In this single-subject meta-analysis, we extracted activation peaks from each subject in each contrast, and we considered individual contrasts to act as standalone studies. The technical details and results of the subject-level analysis are found in Appendix 1, and the results are shown in Figure 1—figure supplement 4 and Figure 1—figure supplement 5. We observe that for most subjects the rostrocaudal principal gradient is spatially determined in both the right and left LPFC hemispheres, albeit the right LPFC shows stronger correspondence across subjects (Figure 1—figure supplement 4). By assessing spatial correspondence of the subject-level gradient with the literature-driven principal LPFC gradient, we observe moderate-to-strong correlation in both LPFC hemispheres (Figure 1—figure supplement 5). We write in Appendix 1, third section:

“Third, to show that the principal LPFC gradient is a not mere reflection of spatial smoothing arising from peak location uncertainty across studies, we reinfer the gradient at the subject-level (Figure 1—figure Supplement 4 and Figure 1—figure Supplement 5) using activation data from the Individual Brain Charting (IBC) dataset (Pinho et al., 2020). The IBC subjects underwent an extensive set of tasks ranging from simple motor tasks to long-term memory and social cognition tasks. The single-subject principal LPFC gradients are shown in Figure 1—figure Supplement 4. These gradients are clearly different from each other and from the literature-inferred gradient, exhibiting more granularity and abrupt transitions than the latter. However, Figure 1— figure Supplement 5 shows a moderate-to-strong correlation between most single-subject gradients and the literature gradient in both LPFC hemispheres. This indicates that the gross spatial layout of principal LPFC gradient is spatially determined at the single subject-level.”

(7) Related to the above point is the clear spatial anchoring of the gradients. Are similar gradients also observed when e.g. focusing only on meta-analytical co-activation of longer-range connections? Systematically varying the connectivity distance thresholds could be something quite meaningful here, as it may reduce potential spatial autocorrelation in observed networks.

We agree that spatial autocorrelation in coactivation patterns among LPFC regions may drive the general layout of the gradients. This view is supported by findings showing that regions more strongly connect to their direct neighbors than to distant regions [4]. For this purpose, we have reinferred the meta-analytic connectivity matrix for 3 Euclidean distances thresholds (20 mm, 40mm, 60 mm), in addition to a case where intra-LPFC coactivations are totally excluded. The results of this analysis are shown in Figure 1—figure supplement 3 and described in the third sub-section of the Results section. Here, we observe that the spatial layout of the principal LPFC does not drastically change when coactivation distances vary, nor when intra-LPFC coactivations are excluded. This indicates that the observed gradients are reproducible for increasingly longrange coactivations, indication a low possibility of biased results due to spatial autocorrelation. We write in Appendix 1, second section:

“Second, to account for the possibility of bias due to spatial auto-correlation among nearby LPFC regions, we re-perform the meta-analysis using longer coactivation distances between regions. The coactivation distance is the Euclidean distance between the centers of mass of each region. The results are shown in Figure 1—figure Supplement 3 and reveal that although the shape of principal LPFC gradient at different coactivation distances (20 mm, 40 mm, and 60 mm) varies, its general rostrocaudal profile is preserved. This suggests that spatial auto-correlation among nearby regions does not induce systematic bias to the overall profile of the principal gradient”.

(8) The analysis on page 6 had a slight suboptimal flavour, as network participation was equated with abstraction. In my view, the subsequent definition of abstraction via cognitive terms is better, so I would possibly rephrase the section and avoid reverse inference (e.g. more dmn, more abstraction).

We agree that the coactivation patterns analysis equates coactivation with Default Mode Network with abstraction. This might give the impression of an informal reverse inference. At first, we decided to abide by the convention used in the literature to describe the DMN’s engagement as necessitating abstraction in function (as opposed to unimodal concrete features), such as in D. Margulies et al., 2016 paper. However, since we agree that this might constitute an informal reverse inference in the context of the current study, we adopted the terms “unimodal” and “transmodal” instead of “concrete” and “abstract” when referring to the coactivation patterns of quintile bins in the revised manuscript. This nomenclature has been used in the gradient-mapping literature, notably in the seminal papers by M. Mesulam 1998 [8] and D. Margulies et al., 2016 [4]. In particular, we modified the title of coactivation analysis Results section to make it:

“Coactivation patterns along the rostrocaudal axis of the LPFC follow a unimodal-totransmodal organization“. Moreover, we modified the caption of Figure 3 to state:

“The coactivation patterns of quintile bins along the principal gradient in the LPFC capture a unimodal-to-transmodal spatial layout in brain network connectivity.”.

(9) The inter-hemispheric asymmetry analysis is interesting and novel. One question that one may want to ask is how far the underlying image processing of the included studies, as well as the meta-analytical synthesis, accounted for potential asymmetries in the brain. In my understanding, a requirement for asymmetry analysis is the mapping of imaging data to a hemisphere unbiased (ie symmetric) template. Is such a step ensured in the included studies and in the meta-analysis inference itself? For example, when carrying out analyses in surface space (results are shown on surfaces) the surface vertices need to ideally be matched across hemispheres to avoid mismatch.

Indeed a symmetric MNI template is essential to analyses that assess inter-hemispheric functional asymmetries in the brain. Although we did not mention it explicitly in the original manuscript, we project the peak coordinates and anatomical masks onto a symmetric MNI template. The template will be available with the data and code to all readers to use for reproducing the results independently. In addition, we have added a clear statement in the methods section describing the use of a symmetric MNI template, and that all of our analysis was carried in volume space not surfaces. Surfaces were only used for the purpose of visualization. Nevertheless, there are certain caveats that are worth mentioning. First, it is not possible to know the exact template used in each of the 14,371 studies, due to the automatic nature of study compiling and peak extraction. Therefore, it is highly likely that some studies have adopted an asymmetric MNI template to normalize the images across subjects before reporting peak coordinates. However, we believe that accounting for spatial uncertainty of peak location via the MKDA technique (radius = 10 mm) reduces possible biases due to template difference across the entire database. Moreover, the units of analysis in the inter-hemispheric asymmetry experiment are quintile bins along the principal gradient in each hemisphere rather than individual voxels or pre-defined anatomical landmarks. Therefore, we believe that ideally matched voxels across all studies, if attainable, will not profoundly change the obtained results.

(10) Wouldn't an additional bootstrap across the studies that were included be useful in order to further model uncertainty across studies (e.g. with respect to the variance explained and gradient shape for Figure 1, topic associations in Figure 4, and assymetries in Figure 5).

We agree that a bootstrap analysis would be informative for quantifying the variability of results across sub-samples of the Neurosynth database. For this reason, we have re-performed some parts of the study on 5000 random sub-samples of the database to estimate the variation in the percent variance explained by the gradients, the change in the shape of the principal gradient, as well as the variance in topic associations and hemisphere asymmetries over the sub-samples. In this analysis, each random sub-sample represented 60% of the original Neurosynth database. The results are shown in Figure 1—figure supplement 1, Figure 1—figure supplement 2, and Figure 4, and Figure 5. Overall, the variations of different measures and metrics are relatively small especially with regards to the shape of the principal gradient and the percentage of variance explained by diffusion embedding components. Collectively, these results show that the choice of studies (given a large number of studies) does not profoundly influence the various outcomes of this study and that the identification and characterization of the principal LPFC gradient are robust. We write in Appendix 1, first section:

“First, to assess the variation in the resultant principal gradient when choice of studies varies, we create 5000 random sub-samples of the Neurosynth dataset and re-run the above analysis on each one. The results are summarized in Figure 1—figure Supplement 1 and Figure 1—figure Supplement 2. Figure 1—figure Supplement 1 shows that most LPFC regions in both hemispheres are assigned to the same position (i.e. quintile bin) along the principal gradient across the majority of sub-samples. The results shown in Figure 1—figure Supplement 2 show that the percentage of variance explained by diffusion embedding components does not vary significantly across the subsamples.”

References

[1] Derek Evan Nee. Integrative frontal-parietal dynamics supporting cognitive control. *eLife*, 10:e57244, 2021

[2] David Badre and Mark D’esposito. Is the rostro-caudal axis of the frontal lobe hierarchical? Nature Reviews Neuroscience, 10(9):659–669, 2009.

[3] Derek Evan Nee and Mark D’Esposito. The hierarchical organization of the lateral prefrontal cortex. *eLife*, 5:e12112, 2016.

[4] Margulies, D. S. *et al.* (2016). Situating the default-mode network along a principal gradient of macroscale cortical organization. *Proc Natl Acad Sci USA* 113, 12574–12579.

[5] Zhang, J., Abiose, O., Katsumi, Y., Touroutoglou, A., Dickerson, B. C., and Barrett, L. F. (2019). Intrinsic functional connectivity is organized as three interdependent gradients. *Scientific reports*, *9*(1), 1-14.

[6] Xu, T., Opitz, A., Craddock, R. C., Wright, M. J., Zuo, X. N., and Milham, M. P. (2016). Assessing variations in areal organization for the intrinsic brain: from fingerprints to reliability. *Cerebral Cortex*, *26*(11), 4192-4211.

[7] Pinho, A. L., Amadon, A., Gauthier, B., Clairis, N., Knops, A., Genon, S., … and Thirion, B. (2020). Individual Brain Charting dataset extension, second release of high-resolution fMRI data for cognitive mapping. *Scientific Data*, *7*(1), 1-16.

[8] Mesulam, M. M. (1998). From sensation to cognition. *Brain: a journal of neurology*, *121*(6), 1013-1052